# Two-dimensional multiferroic material of metallic p-doped SnSe

Ruofan Du[1,3], Yuzhu Wang[1,3], Mo Cheng[1], Peng Wang[1], Hui Li[1], Wang Feng[1], Luying Song[1], Jianping Shi [1] ✉ & Jun He [2] ✉

Two-dimensional multiferroic materials have garnered broad interests attributed to their magnetoelectric properties and multifunctional applications. Multiferroic heterostructures have been realized, nevertheless, the direct coupling between ferroelectric and ferromagnetic order in a single material still remains challenging, especially for two-dimensional materials. Here, we develop a physical vapor deposition approach to synthesize two-dimensional p-doped SnSe. The local phase segregation of $SnSe_2$ microdomains and accompanying interfacial charge transfer results in the emergence of degenerate semiconductor and metallic feature in SnSe. Intriguingly, the room-temperature ferrimagnetism has been demonstrated in two-dimensional p-doped SnSe with the Curie temperature approaching to ~337 K. Meanwhile, the ferroelectricity is maintained even under the depolarizing field introduced by $SnSe_2$. The coexistence of ferrimagnetism and ferroelectricity in two-dimensional p-doped SnSe verifies its multiferroic feature. This work presents a significant advance for exploring the magnetoelectric coupling in two-dimensional limit and constructing high-performance logic devices to extend Moore's law.

The multiferroic materials, those simultaneously possess ferromagnetic and ferroelectric orders, have attracted increasing attentions because of their emerging physical properties (e.g., magnetoelectric coupling[1–3], nonreciprocity[4,5], topological order[6], and thermal Hall effect[7], etc.) and multifunctional applications in memory device[8], spintronic device[9], and nondestructive data storage[10]. In general, the electron spin order in partially filled $d/f$ orbitals of transition metals results in the evolution of magnetism and breaks the time-reversal symmetry. However, the ferroelectricity often derives from the residual polarization due to the stable off-centered ion with empty $d/f$ orbitals, which breaks the space-inversion symmetry[11]. In this regard, the exploration of multiferroic in low symmetry materials is promising and considerable efforts have also been made (e.g., $NiI_2$[1], $GaFeO_3$[12], $BiFeO_3$[13], $TbMnO_3$[14], and $MnWO_4$[15], etc). Although two-dimensional (2D) multiferroic materials have long been sought for constructing high-performance magnetoelectric coupling devices, the progress is still unsatisfactory.

Group-IV monochalcogenides, such as SnSe, possess the distorted puckered structure[16,17] and have been predicted to be ferroelastic-ferroelectric multiferroics with spontaneous electric polarization and lattice strain[18,19]. Recently, the ferroelectricity has been experimentally validated in SnSe and SnS, respectively. For example, the shape-dependent in-plane piezoelectric response (with a piezoelectricity of ~19.9 pm $V^{-1}$) was observed in the solvent assistance synthesized SnSe nanowall/microsphere[20]. The room-temperature ferroelectricity was found to exist in physical vapor deposition (PVD) grown SnS nanosheet with a thickness below 15 layers[21]. The robust ferroelectricity with a critical temperature close to ~400 K was discovered in molecular beam epitaxy (MBE) synthesized monolayer SnSe on graphene, and its ferroelectric domain could be manipulated by the bias voltage[22]. Notably, the ferromagnetism exploration of group-IV monochalcogenides is absent, although the ferromagnetic-ferroelectric multiferroics are promising for constructing magnetoelectric devices with efficient

[1]The Institute for Advanced Studies, Wuhan University, 430072 Wuhan, China. [2]Key Laboratory of Artificial Micro- and Nano-structures of Ministry of Education, School of Physics and Technology, Wuhan University, 430072 Wuhan, China. [3]These authors contributed equally: Ruofan Du, Yuzhu Wang. ✉e-mail: jianpingshi@whu.edu.cn; He-jun@whu.edu.cn

writing and lower energy-cost reading[8]. As a typical member, SnSe is a nonmagnetic semiconductor, nevertheless, the magnetic moment can be induced through the hole doping[23,24].

Here, we design a high-throughput PVD method to synthesize 2D p-doped SnSe on mica in a controlled manner. A metallic feature is uncovered in 2D p-doped SnSe due to the local phase segregation of $SnSe_2$ microdomains and accompanying interfacial charge transfer. The room-temperature ferrimagnetism and robust ferroelectricity are found to exist in PVD-synthesized 2D p-doped SnSe simultaneously, highly suggestive of its multiferroic feature. By combining density functional theory (DFT) calculations and electrical transport/piezo-response force microscopy (PFM) measurements, the internal mechanism is clarified unambiguously.

## Results

### Controlled synthesize thickness-tunable SnSe nanosheets

2D SnSe nanosheets are successfully synthesized on mica substrates by using an atmospheric pressure PVD method with SnSe powders as the precursors. The schematic diagram of the growth process and crystal structure of SnSe along *a*-axis is depicted in Fig. 1a. The orthorhombic structure is distinguished and the high Grüneisen parameter for such a structure results in anharmonic and anisotropic bonding. The freshly cleaved fluorphlogopite mica $(KMg_3(AlSi_3O_{10})F_2)$ is selected as the growth substrate in view of its chemically inert feature and atomically smooth surface, as well as the weak van der Waal interaction with precursors that allows them to migrate with relatively low barrier, which is critical for the epitaxy growth of 2D materials. Other substrates (*e.g.*, Au foil, soda-lime glass, $SiO_2/Si$, and sapphire) are also used to synthesize SnSe, nevertheless, smaller and thicker nanosheets

are obtained (Supplementary Fig. S1), reconfirming the advantage of mica substrate for growing large-domain and ultrathin SnSe. X-ray diffraction (XRD) measurements were then performed on transferred samples to identify the phase structure of SnSe (Supplementary Fig. S2). Three main diffraction peaks at ~15.4°, ~31.1°, and ~64.8° are assigned as (200), (400), and (800) planes of SnSe, respectively, according to the JCPDS card no. 48-1224, indicating the orthorhombic phase and layered structure along the *c*-axis. Figure 1b displays the Raman spectrum of as-grown SnSe, the characteristic peaks at ~33.2, ~70.6, ~109.9, ~131.5, and ~150.0 $cm^{-1}$ correspond to $B^1_{3g}$, $A^1_g$, $B^2_{3g}$, $A^2_g$, and $A^3_g$ modes of SnSe, respectively. The corresponding Raman intensity mappings of $A^1_g$ and $B^2_{3g}$ modes for a tetragonal SnSe nanosheet manifest a rather uniform color contrast, suggestive of its high thickness uniformity (Fig. 1c and Supplementary Fig. S3).

Optical microscopy (OM) and atomic force microscopy (AFM) measurements were performed on as-grown samples to evaluate the morphology, domain size, and thickness evolution of 2D SnSe with the precursor–substrate distance. Apparently, the average edge lengths and thicknesses of SnSe nanosheets are tunable from ~17.3 to ~61.4 μm and from ~3.4 to ~116.4 nm, respectively, with increasing the precursor-substrate distance from 12 to 16 cm (Fig. 1d–k and Supplementary Fig. S4). Interestingly, tetragonal SnSe nanosheets with an average edge length of ~61.4 μm are synthesized as the distance is set to be 12 cm (Fig. 1d, g), nevertheless, circle SnSe nanosheets (with an average edge length of ~17.3 μm) are evolved as the distance increased to 16 cm (Fig. 1f, i). The coexisting tetragonal and circle SnSe nanosheets are obtained at a distance of 14 cm, with the corresponding average edge lengths of ~29.7 and ~28.3 μm, respectively, as shown in Fig. 1e, h. Notably, the circle SnSe nanosheets possess a much thinner thickness

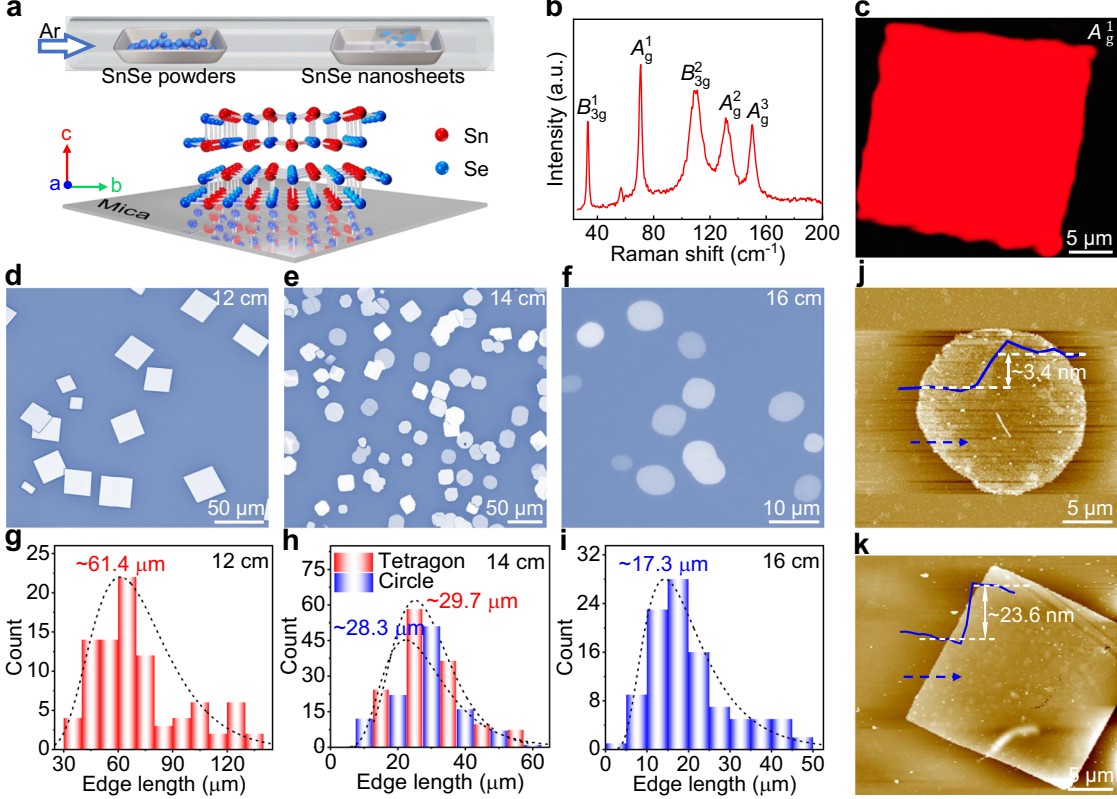

**Fig. 1 | Controllable synthesis of thickness-tunable SnSe nanosheets on mica. a** Schematic diagram of the growth process and crystal structure of SnSe on mica. **b** Raman spectrum of as-grown SnSe on mica. **c** Raman intensity mapping image of $A^1_g$ mode for a tetragonal SnSe nanosheet, showing its thickness uniformity. **d**–**f** OM images of as-grown SnSe that synthesized at different precursor-substrate

distances of 12, 14, and 16 cm, respectively, revealing variable domain sizes and morphologies. **g**–**i** Corresponding edge length distributions. **j**, **k** AFM images and corresponding height profiles analyses of tetragonal and circle SnSe nanosheets on mica.

than those of tetragonal counterparts, as confirmed by the AFM results in Fig. 1j, k. The decreased SnSe vapor concentration with increasing the precursor-substrate distance results in a limited growth rate and kinetics. A similar growth behavior was also demonstrated in PVD synthesis of SnS on mica[21]. In brief, domain size and thickness-tunable SnSe nanosheets have been synthesized, which provides an ideal platform for exploring exotic physical properties (e.g., ferroelectricity and ferromagnetism) and developing multifunctional applications in electronic devices.

### Atomic structure of PVD-synthesized 2D SnSe

To determine the atomic structure and crystalline quality of PVD-synthesized 2D SnSe nanosheets, high-resolution transmission electron microscopy (HRTEM) and high-angle annular dark-field scanning transmission electron microscopy (HAADF-STEM) measurements were performed on transferred samples. Figure 2a reveals the low-magnification TEM image of a well-shaped tetragonal SnSe nanosheet and its high crystalline quality is convinced by the regular morphology. A series of selected area electron diffraction (SAED) patterns captured from four random positions of such a SnSe nanosheet are shown in Fig. 2b. Only one set of arranged diffraction spots is observed, almost the same orientation verifies the single crystalline property and high crystallinity of PVD-synthesized 2D SnSe nanosheets.

In addition, energy dispersive X-ray spectroscopy (EDS) measurements were carried out to identify the chemical constitutions and their distributions (Fig. 2c–f). The uniform color contrast within the nanosheet shows the high crystalline quality of 2D SnSe. The typical atomic-resolution HAADF-STEM images and corresponding fast Fourier transform (FFT) pattern of tetragonal SnSe are depicted in Fig. 2g, h and Supplementary Fig. S5, respectively, a perfect lattice is obviously

observed to show almost no visible defect, in line with the simulative result in Fig. 2i, suggestive of the high crystalline quality of 2D SnSe. The lattice constant is measured to be ~2.1 Å, consistent with the (020) plane spacing of SnSe (Supplementary Fig. S6). The AA stacking order of SnSe is obviously observed from the atomic-resolution HAADF-STEM result, where the top layer is precisely matched with the bottom layer in the $a$–$b$ plane, as described by the atomic structure model in Fig. 2i. Remarkably, this stacking configuration possesses a stable ferroelectric phase, which has been clarified by the calculated result[25]. The atomic structure of circle SnSe is also detected and shown in Supplementary Fig. S7, where the same atomic arrangement with tetragonal SnSe is clearly obtained. Nevertheless, the relatively high density of defect is observed due to its low crystalline quality, and a similar phenomenon is also demonstrated in PVD-synthesized 2D circle SnS[21].

### The metallic feature of PVD-synthesized 2D p-doped SnSe

To investigate the electronic property and its anisotropy of PVD-synthesized 2D SnSe, a series of back-gate devices were thus constructed by transferring samples on Au electrodes, with the OM image shown in Fig. 3a. Notably, the electrode directions of 1–2 and 3–4 are identified as zigzag (ZZ) and armchair (AC) orientations of SnSe, respectively. The output characteristic curves (drain–source current ($I_{ds}$) $vs$ drain–source voltage ($V_{ds}$)) collected both along 1–2 and 3–4 directions show nearly linear and symmetric relationships under different back-gate voltages ($V_{gs}$) from −60 to +60 V, indicating the Ohmic-type contact between SnSe nanosheets and Au electrodes. Furthermore, the $I_{ds}$ values remain invariant with the $V_{gs}$ changing from −60 to +60 V, suggesting the metallic feature of SnSe (Fig. 3b), in contrast to intrinsically semiconducting behavior[26]. Interestingly, the much higher conductivity is clearly observed along the ZZ orientation

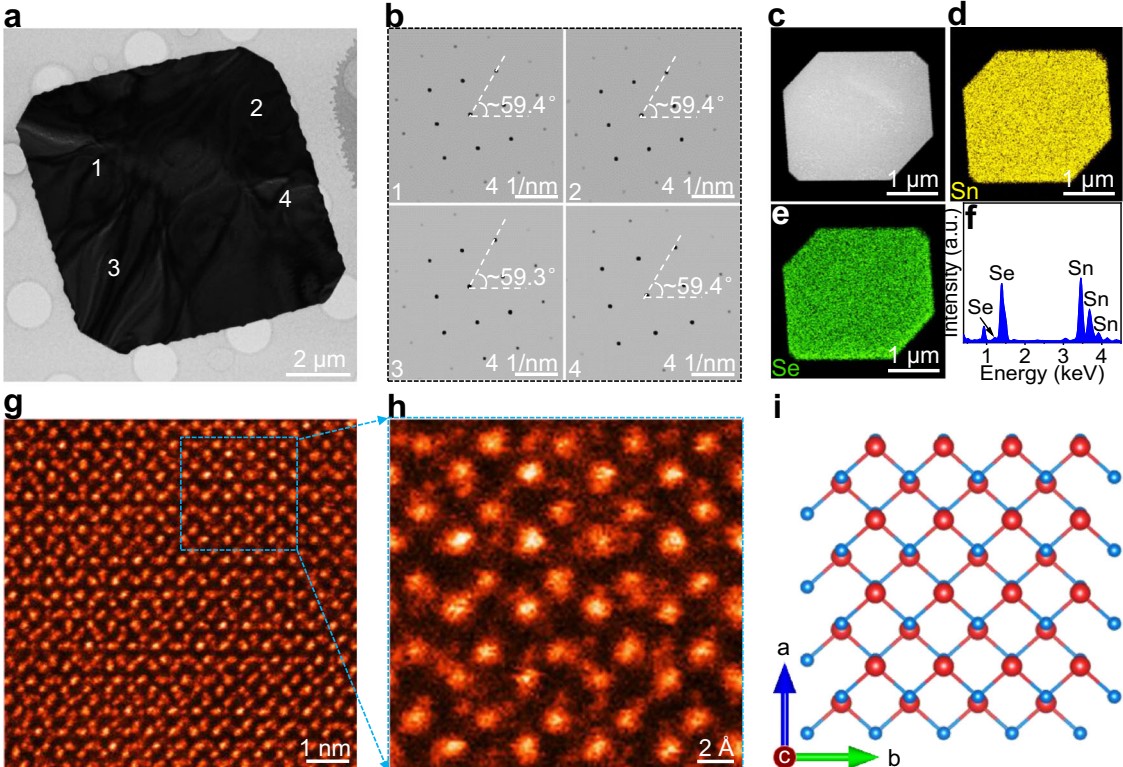

**Fig. 2 | Atomic structure of PVD-synthesized 2D SnSe. a** Low-magnification TEM image of a typical tetragonal SnSe nanosheet. **b** Corresponding SAED patterns captured from the random positions labeled with numbers 1–4 in **a**. **c** Low-magnification TEM image of a tetragonal SnSe nanosheet. **d, e** Corresponding EDS mapping images of Sn and Se, respectively, showing the uniform element distribution. **f** Quantified analysis of the EDS result. **g** Atomic-resolution HAADF-STEM image of a tetragonal SnSe nanosheet. **h** Zoomed-in HAADF-STEM image. **i** Atomic structure model of tetragonal SnSe viewed along the $c$-axis.

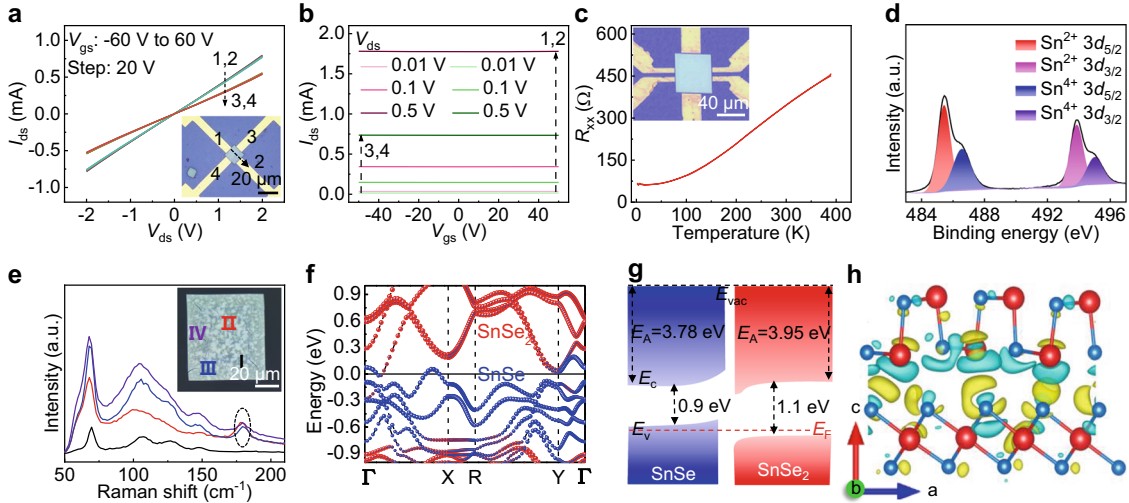

**Fig. 3 | Metallic behavior of PVD-synthesized 2D p-doped SnSe. a** Output characteristic curves of SnSe, collected both along 1–2 and 3–4 directions. Inset is the corresponding OM image of a back-gate device. **b** Transfer characteristic curves of the SnSe back-gate device. **c** Temperature-dependent longitudinal resistance of SnSe (with the thickness of ~46 nm). Inset is the corresponding OM image of a Hall bar device. **d** XPS spectrum of transferred 2D p-doped SnSe nanosheets on SiO$_2$/Si, showing the coexistence of Sn$^{2+}$ and Sn$^{4+}$. **e** Raman spectra captured from four random positions in Ar plasma treated sample. The characteristic peaks of SnSe$_2$ are indicated by the black circle. Inset is the corresponding OM image. **f, g** Calculated band structure and energy band diagram of SnSe/SnSe$_2$. **h** Differential charge density of SnSe/SnSe$_2$ (isosurface value of 0.001 e/A$^3$). Yellow and blue isosurface contours represent the charge accumulation and depletion, respectively.

(~9.76 × 10$^3$ S m$^{-1}$) than that of the AC counterpart (~6.72 × 10$^3$ S m$^{-1}$), manifesting the anisotropic electronic property of SnSe (Supplementary Fig. S8). The distinctive crystal structure of SnSe determines such an unusual phenomenon, as has been demonstrated in bulk SnSe[27]. The metallic feature of PVD-synthesized 2D SnSe is also confirmed by the reduced longitudinal resistance ($R_{xx}$) with decreasing the temperature from ~390 to ~2 K (Fig. 3c), as well as the intensive density of states (DOS) near the Fermi level (Supplementary Fig. S9). In addition, the carrier concentration of 2D p-doped SnSe is extracted to be ~8.31 × 10$^{19}$ cm$^{-3}$ at 150 K and ~5.70 × 10$^{19}$ cm$^{-3}$ at 2 K, by performing the Hall measurements (Supplementary Fig. S10).

By precisely controlling the cooling rate during the sample growth process, SnSe$_2$ microdomains are evolved within SnSe, which results in a degenerate semiconductor and therefore a metallic behavior[24]. In consideration of different thermal conductivities of SnSe (~0.69 W m$^{-1}$ K$^{-1}$)[16] and mica substrate (~0.183 W m$^{-1}$ K$^{-1}$)[28], SnSe$_2$ microdomains could be segregated during the growth process. Furthermore, the similar formation enthalpy between SnSe$_2$ (−0.43 eV/atom) and SnSe (−0.56 eV/atom) will be simple to induce the secondary phase (SnSe$_2$) in the host matrix phase (SnSe) or Sn vacancies in SnSe[29]. To confirm this speculation, X-ray photoemission spectroscopy (XPS) measurements were thus performed to characterize the chemical states of PVD-synthesized SnSe, with the results shown in Fig. 3d and Supplementary Fig. S11. The binding energies of ~485.4 and ~493.9 eV are attributed to Sn$^{2+}$ 3$d_{5/2}$ and 3$d_{3/2}$, respectively. Interestingly, additional two characteristic peaks at ~486.6 and ~495.0 eV are obviously observed, which are assigned to Sn$^{4+}$ 3$d_{5/2}$ and 3$d_{3/2}$, respectively, highly suggestive of the generation of Sn$^{4+}$ state in the as-grown sample. Additionally, the Ar plasma treatments and Raman characterizations were also implemented to determine the existence of SnSe$_2$ microdomains (Fig. 3e and Supplementary Fig. S12). Besides characteristic Raman peaks of SnSe at ~70, ~108, ~132, and ~150 cm$^{-1}$, a new peak (~180 cm$^{-1}$) is obtained, which corresponds to the $A_{1g}$ vibration mode of SnSe$_2$, indicative of the formation of SnSe$_2$ microdomains. To further confirm the formation of SnSe$_2$ microdomains in SnSe, TEM measurements were performed and hexagonal SnSe$_2$ microdomains with the domain size changing from ~23 to ~105 nm and average coverage of ~15.6% were obviously observed

(Supplementary Figs. S13, 14). The XPS depth analysis was also carried out on SnSe/SnSe$_2$ heterostructure (with a thickness of ~16 nm) to evaluate the thickness distribution of SnSe$_2$ microdomains, and their thicknesses were determined to be 2–4 nm (Supplementary Fig. S15).

DFT calculations were thus carried out to clarify the metallic feature of PVD-synthesized 2D SnSe. To minimize the lattice mismatch between the stacking layers, a rectangular unit cell of SnSe$_2$ is constructed. And the SnSe/SnSe$_2$ heterostructure is built by stacking 1 × 3 supercell of SnSe on 1 × 2 supercell of SnSe$_2$. The optimized in-plane lattice constants of monolayer SnSe$_2$ are $a$ = 3.87 Å, $b$ = 6.70 Å, while the corresponding in-plane lattice constants of monolayer SnSe are $a$ = 4.29 Å, $b$ = 4.40 Å. Notably, the similar lattice stretching of hexagonal transition-metal dichalcogenides (TMDCs) is also applied in the analogous heterostructure, such as black phosphorous/MoS$_2$[30]. The detailed crystal structures of SnSe$_2$, SnSe, and SnSe/SnSe$_2$ heterostructure are presented in Supplementary Fig. S16 and Table S1. The band structures of SnSe/SnSe$_2$ calculated with different methods are presented in Fig. 3f and Supplementary Fig. S17, respectively, and the Fermi level is crossed by several bands, indicative of its metallic feature, echoing well with the experimental findings. Meanwhile, the electron transfer from SnSe to SnSe$_2$ is also confirmed both by the energy band diagram and charge distribution in Fig. 3g, h, which convinces the formation of p-doped SnSe. In short, 2D metallic p-doped SnSe with high conductivity is confirmed because of the formation of SnSe$_2$ microdomains, which offers a playground for exploring quantum phenomena, such as weak antilocalization effect (Supplementary Fig. S18), and multifunctional applications in electronic device and energy-related field.

## The ferrimagnetism in PVD-synthesized 2D p-doped SnSe

2D multiferroic materials have attracted intensive interest due to their unique physical properties and multifunctional applications[1]. The ferroelectricity has been theoretically predicted and experimentally investigated in SnSe nanowalls[20] and monolayer films[22]. Nevertheless, the ferromagnetism exploration of 2D SnSe is still absent. Superconducting quantum interference device (SQUID) measurements were performed on transferred p-doped SnSe nanosheets (with an average thickness of ~41.4 nm) on SiO$_2$/Si to determine the intrinsic magnetism. Zero-field-cooled (ZFC) and field-cooled (FC) magnetization curves

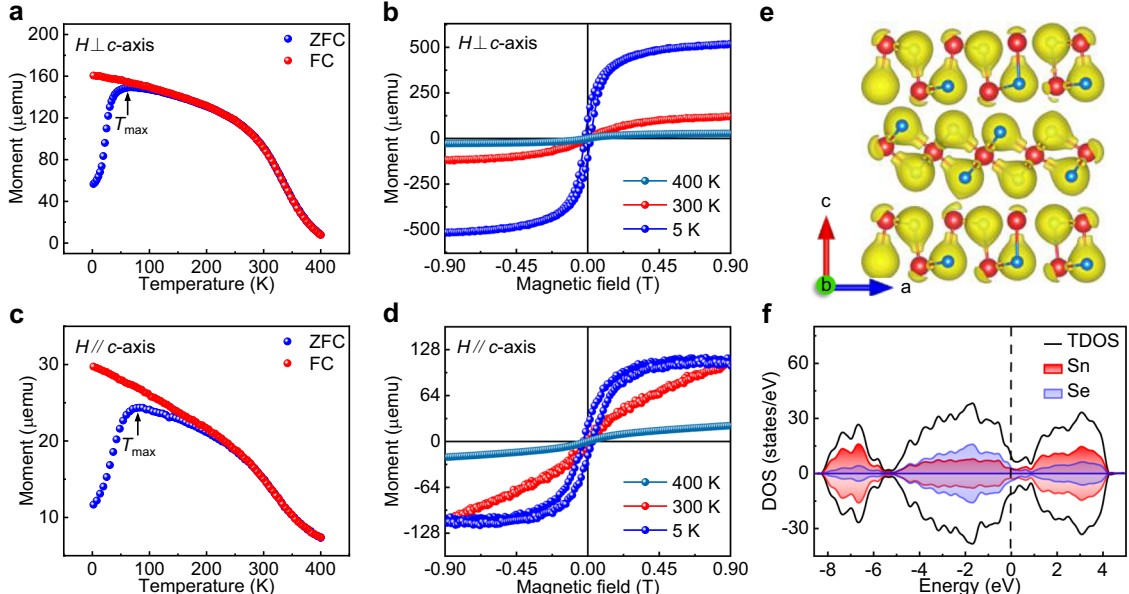

**Fig. 4 | Ferrimagnetism of PVD-synthesized 2D p-doped SnSe. a, c** Temperature-dependent magnetic moment of 2D p-doped SnSe nanosheets with a vertical and parallel magnetic field at 100 Oe. $T_{max}$ is defined as the temperature at which the maximum magnetic moment is obtained during the ZFC process. **b, d** Magnetic hysteresis loops of 2D p-doped SnSe at ~5, ~300, and ~400 K with the magnetic field vertical and parallel to the $c$-axis, respectively. **e** Differential charge density of SnSe/SnSe$_2$ (isosurface value of 0.03705 e Bohr$^{-3}$). Yellow isosurface contours represent the charge accumulation. The same direction of electron indicates the appearance of ferromagnetism. **f** Projected spin-polarized DOS of SnSe/SnSe$_2$. The Fermi level is set as zero.

were collected with the magnetic field (100 Oe) vertical and parallel to the $c$-axis, respectively (Fig. 4a, c). Interestingly, both ZFC−FC curves present ferrimagnetic features with the Curie temperature ($T_c$) up to ~337 K (Supplementary Fig. S19), consistent with the theoretically predicted value (~325 K)[31]. The maximum magnetic moments are obtained for ZFC curves at the temperatures of ~60 K (magnetic field vertical to $c$-axis) and ~75 K (magnetic field parallel to $c$-axis), and such a phenomenon is also observed in spin-glass[32] and superparamagnetic material[33].

The magnetic hysteresis loops of 2D p-doped SnSe nanosheets are clearly achieved at the temperatures of ~5, ~300, and ~400 K under the parallel and vertical magnetic fields (Fig. 4b, d), confirming the long-range ferrimagnetic order. Notably, a relatively low remanence is obtained in 2D p-doped SnSe, suggesting weak permanent magnetism and ferrimagnetism. Even so, the remanences are still observed even at ~300 K, suggestive of the room-temperature ferrimagnetism in 2D p-doped SnSe. The corresponding remanence, coercivity, and saturation magnetism are extracted and shown in Table S2. To further trace the magnetic order evolution of 2D p-doped SnSe, the temperature-dependent magnetic moment measurements (from ~2 to ~700 K) were performed, and the ferromagnetic order was disappeared at ~665 K, reconfirming the high-temperature ferrimagnetic property of 2D p-doped SnSe (Supplementary Fig. S20). The high temperature long-range ferromagnetic order in 2D p-doped SnSe can be explained by the kinetic exchange mechanism[34]. In detail, during the charge transfer process, the parallel spin alignment contributes to the hopping between the adjacent states, which reduces the kinetic energy and stabilizes the long-range ferromagnetic order at high temperatures. Besides, in view of the abundant interfaces between SnSe and SnSe$_2$, the exchange interactions of magnetic moments at the interfaces are enhanced and contribute to the emergence of long-range ferromagnetic order in 2D p-doped SnSe. The easy axis is assigned to the in-plane due to its higher saturation magnetization and coercivity than those of the out-of-plane (Table S2). To further determine the magnetic domain and switching behavior, the magnetic force microscope (MFM) measurements were then performed on the transferred p-doped SnSe with different thicknesses at room temperature.

Interestingly, the phase deviation between p-doped SnSe and non-magnetic SiO$_2$/Si substrate is obviously observed, indicating its ferromagnetic order (Supplementary Fig. S21), the similar phenomena are also demonstrated in monolayer V-doped WSe$_2$[35] and monolayer VSe$_2$[36]. Notably, the single magnetic domain is obtained for p-doped SnSe, which is similar to 2D ferromagnetism Fe$_3$GaTe$_2$[37]. However, the switching behavior between the ferromagnetic and paramagnetic order is not observed, possibly due to the weak magnetoelectric coupling in 2D p-doped SnSe.

DFT calculations were then performed to provide further insights on the ferromagnetism in 2D p-doped SnSe. The physical origin of magnetism can be understood by the Stoner mechanism. In detail, the DOS at the Fermi energy ($D_{(Ef)}$ = 9.35 states/eV) and the exchange interaction strength ($J$ = 0.20 eV) are calculated for 2D p-doped SnSe (the detailed calculation process is presented in the method section), and the "Stoner Criterion" ($D_{(Ef)}J = 1.87 > 1$) is found to be well-satisfied, suggesting that the exchange interaction is larger than the loss of kinetic energy and thus induces the formation of ferromagnetism[38]. The obtained value of $J$ is comparable with those of 2D ferromagnetic FeTe (0.30−0.40 eV)[39] and ferromagnetic transition-metal halides (0.20−0.57 eV)[40]. Such a large exchange interaction strength in 2D p-doped SnSe indicates the strong interaction and coupling between the magnetic moments, which is favorable for stabilizing the long-range ferromagnetism. Moreover, the reduced longitudinal resistance with decreasing the temperature (Fig. 3c) and the appearance of positive magnetoresistance throughout the whole temperature range (Supplementary Fig. S18) suggest the robust long-range Ruderman−Kittel−Kasuya−Yosida (RKKY) interaction, which is also favorable for stabilizing the long-range ferromagnetic order in 2D p-doped SnSe. Notably, the relatively high $D_{(Ef)}$ in 2D p-doped SnSe may lead to the exchange-field splitting near the VBM and contributes to the formation of ferromagnetism. In addition, the phase segregation of SnSe$_2$ microdomains introduces the strain/lattice distortion, which possibly modulates the Berry curvature diploe and results in the appearance of orbital magnetism[41]. The differential charge density of SnSe/SnSe$_2$ in Fig. 4e shows the same electron orientation, indicating the appearance of a ferrimagnetism state, consistent with the

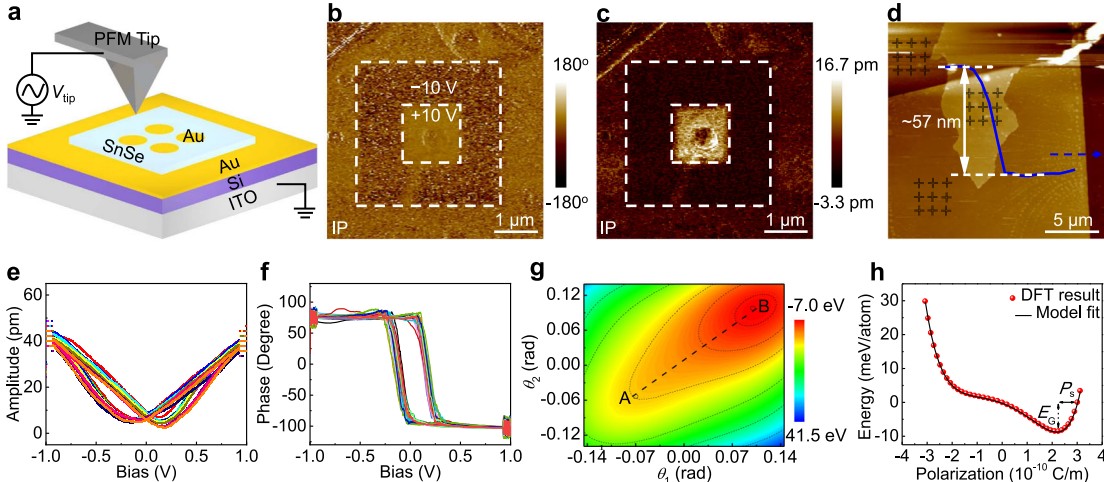

**Fig. 5 | The ferroelectricity determination of 2D p-doped SnSe. a** Schematic diagram of the PFM measurement. **b, c** Phase and amplitude images of 2D p-doped SnSe after poling with ±10 V, showing stable polarization states. **d** AFM image and corresponding height profile analysis of a 57-nm-thick tetragonal SnSe nanosheet with a top Au electrode. **e, f** Corresponding PFM amplitude hysteresis loops and phases captured from the selected positions labeled in **d**. **g** Free-energy contour plot of SnSe/SnSe$_2$ interface according to the tilting angles ($\theta_1$ and $\theta_2$). The para-electric phase (A) and ferroelectric phase (B) are marked. **h** Polarization-dependent single-well potential. $E_G$ and $P_s$ are defined as the ground-state energy (potential barrier) and spontaneous polarization, respectively.

experimental results. The projected spin-polarized DOS in Fig. 4f reveals that the states around the Fermi level are determined by Se atoms, and the asymmetric DOS of two spin channels results in a total magnetic moment of -0.7881 $\mu_B$, much higher than that of intrinsic SnSe (-0.5175 $\mu_B$), indicative of the emergence of ferrimagnetism in PVD-synthesized 2D p-doped SnSe nanosheets. The evolution of fer-rimagnetism in 2D SnSe has laid a solid foundation for constructing high-performance spintronic devices.

## The ferroelectricity in PVD-synthesized 2D p-doped SnSe

PFM is a noninvasive and powerful technology for determining ferroelectricity and thus is used for characterizing the transferred 2D p-doped SnSe nanosheets on Au/Si/ITO at room temperature, with the schematic diagram shown in Fig. 5a. To eliminate the electrostatic effect between PFM tip and surface charge under high external electric field[42], the top Au electrodes were deposited on 2D p-doped SnSe nanosheets to avoid the local charge/ion accumulation[43], with the OM image presented in Supplementary Fig. S22. Figure 5b, c reveals the PFM phase and amplitude images captured from 2D p-doped SnSe after writing a box-in-box pattern with the opposite bias (±10 V). Interestingly, the PFM phase image (Fig. 5b) reveals bright and dark regions with a nearly 180° phase contrast, corresponding to the upwards and downwards polariza-tion states, respectively. Meanwhile, an enhanced PFM amplitude signal is obtained in the electrically poled region (Fig. 5c, white dashed square), the clear and well-defined domain wall is observed on the boundary. Such results indicate the robust and intrinsic ferroelectric feature in 2D p-doped SnSe, as well as its switching behavior. The topographic images of 2D p-doped SnSe nanosheets with different thicknesses are shown in Fig. 5d and Supplementary Fig. S23, the corresponding amplitude hysteresis loops and phases are obtained (Fig. 5e, f). The well-defined butterfly loops of ampli-tude signals and the distinct 180° switching of phases reconfirm the robust ferroelectric polarization in 2D p-doped SnSe.

To determine the stacked configuration of top and bottom SnSe nanosheets, DFT calculations were performed with the results shown in Supplementary Fig. S24. Interestingly, the energy of fer-roelectric stack is -3.23 meV lower than that of the anti-ferroelectrically stacked structure, and thus the ferroelectrically staked configuration is established. The large distance between SnSe layers in the SnSe/SnSe$_2$ heterostructure results in a weak interlayer

coupling and possibly induces the relatively small energy difference between the antiferroelectric and ferroelectric phase. In spite of this, such an energy difference is still comparable with that of bilayer SnSe (-2 meV for AA stacking order)[25]. The free energy of the SnSe/SnSe$_2$ interface is calculated under different distortion angles ($\theta$) along the AR direction and the single-well potential is thus obtained by the Landau model fitting (Supplementary Fig. S25). Interestingly, only one lowest energy point is obviously observed, indicating the ferroelectric spontaneous polarization in SnSe/SnSe$_2$. Meanwhile, the free-energy contour of SnSe/SnSe$_2$ is also plotted in Fig. 5g, and the saddle point A corresponds to the paraelectric phase with cen-tral symmetry. The $\theta_1$ and $\theta_2$ are defined as the angles those mea-sured along the AR direction of SnSe (Supplementary Fig. S26). Notably, the relationship between free energy and polarization can be established by calculating the polarization under different dis-tortion angles (Fig. 5h). The $E_G$ is defined as the transfer potential barrier from ferroelectric phase (B) to the paraelectric phase (A), and thus calculated to be -164 meV, much higher than that of other 2D ferroelectric materials (e.g., α-In$_2$Se$_3$[44], CnInP$_2$S$_6$[45,46], and MoTe$_2$[47]). Meanwhile, the spontaneous polarization intensity ($P_s$) of the SnSe/SnSe$_2$ interface is obtained to be -0.58 × 10$^{-10}$ C/m, com-parable to the monolayer SnSe[31], which means that the ferroelec-tricity of SnSe is still maintained even under the depolarizing field introduced by SnSe$_2$. The ferroelectric polarization in 2D p-doped SnSe possibly originates from the strain/lattice distortion and sub-sequent off-center displacement of Sn, due to the phase segregation of SnSe$_2$ microdomains. The different chemical environment of Sn at the interfaces leads to the formation of uncompensated charge transfer and dipole[48,49]. In addition, the corresponding ionic-potential anharmonicity in 2D p-doped SnSe may also contribute to the emergence of ferroelectric polarization[31,50]. The dipole moments of 2D p-doped SnSe are calculated by using the bond-valence method and Debye equation (Supplementary Fig. S27, the detailed calculation process is presented in the method section) and the reduced dipole moment value (-0.0001 Debye) comparing with that of intrinsic SnSe (-0.0510 Debye) suggestive of the weak fer-roelectricity in 2D p-doped SnSe, consistent with the experimental results. Such results suggest that the PVD-synthesized 2D p-doped SnSe is a ferroelectric material, which opens the possibilities for applications in sensors, actuators, and non-volatile memory devices[51].

## Discussion

In summary, large-domain and thickness-tunable SnSe nanosheets have been successfully synthesized on mica substrates by a simple PVD method. Given the similar formation enthalpy between $SnSe_2$ and SnSe, the local phase segregation of $SnSe_2$ microdomains and interfacial charge transfer are discovered, which result in the emergence of degenerate semiconductor and metallic feature in 2D p-doped SnSe. More interestingly, the room-temperature ferrimagnetism and robust ferroelectricity have been verified in PVD-synthesized 2D metallic p-doped SnSe by combining DFT calculations, SQUID, and MFM/PFM measurements, and the multiferroic property is thus established. Such results present a breakthrough toward the controllable synthesis of 2D multiferroic materials, and open up a possibility for future industrial implementation of 2D multiferroic materials in next-generation logic devices.

## Methods

### PVD synthesis of 2D p-doped SnSe

The ambient pressure PVD was conducted to grow 2D p-doped SnSe nanosheets in a dual heating zone furnace equipped with a 1-inch outer diameter quartz tube. The SnSe (~99.999%, 50 mg, Alfa Aesar) precursors were placed in a quartz boat and located in the center of upstream heating zone. The freshly cleaved mica substrates ($10 \times 10 \times 0.2$ mm, TaiYuan Fluorphlogopite Mica Company Ltd.) were located 4 cm from the front of downstream heating zone. Notably, the SnSe powders were covered by molecular sieves to reduce the evaporation rate. Before conducting the PVD growth, the quartz tube was purged by ~500 sccm high-purity Ar gas and kept for ~5 minutes to remove the air and humidity. After that, the temperatures of upstream and downstream heating zone were heated to ~660 and ~430 °C, respectively, within 30 minutes. In addition, ~100 sccm Ar was introduced as the carrier gas during the PVD growth process. The target temperature was kept for 10 minutes. And then, the furnace was cooled down to room temperature naturally.

### Transfer and characterization of 2D p-doped SnSe

The PVD-synthesized 2D p-doped SnSe nanosheets were transferred by using the polystyrene (PS) assistant method[52]. In detail, 13 g of PS particles were dissolved in ~100 mL of toluene. The prepared PS solutions were spin coated onto SnSe/mica and then baked for ~20 minutes at ~60 °C. The edges of PS/SnSe/mica were scraped for the fast separation. Subsequently, the PS/SnSe/mica were floated on the surface of water. When the PS/SnSe was transferred onto target substrates, a baking step (at ~110 °C for ~15 minutes) was carried out to remove the water residues. Finally, the PS/SnSe was soaked in toluene for ~30 minutes to dissolve PS and then blown dry by nitrogen gas. The morphology, domain size, thickness, phase structure, optical property, and crystalline quality of 2D p-doped SnSe nanosheets were characterized by OM (Olympus BX53M), AFM (Dimension Icon, Bruker), XPS (Thermo Scientific Kα + system, and the binding energies were calibrated by C1s at ~284.8 eV), XRD (Rigaku Smartlab SE), Raman spectroscopy (Renishaw, with the excitation light of ~532 nm), and TEM (JEOL JEM-F200, with the acceleration voltage of ~200 kV). The atomic-resolution HAADF-STEM imaging was conducted on an aberration-corrected STEM JEOL ARM-200F with an acceleration voltage of ~80 kV.

### Ferroelectric characterization

PFM measurements were performed using a commercial AFM (Bruker Multimode 8) with a Pt/Ir-coated Si cantilever tip (spring constant: 3 N/m). For the local electric measurements, the bias voltage of ±10 V was applied to the sample.

### Electrical and ferrimagnetism measurements

The PVD-synthesized 2D p-doped SnSe nanosheets with different thicknesses were transferred onto $SiO_2$/Si substrates with pre-

evaporated Au electrodes (~40 nm). The electrical transport measurements were performed under vacuum (<1.3 mTorr) and dark conditions by using a semiconductor characterization system (Keithley 4200-SCS). The ferrimagnetism properties of 2D p-doped SnSe nanosheets were measured by SQUID (Quantum Design, MPMS3) using the reciprocating sample option. The magnetic hysteresis loops were measured by using the max slope position and linear regression fitting parameters to eliminate centering errors at zero moment. Temperature-dependent magnetic moments of 2D p-doped SnSe nanosheets were measured by using ZFC and FC modes with the cooling rate of 3 K min$^{-1}$. MFM measurements were performed using a commercial AFM (Bruker Multimode 8) with a Co-Cr-coated tip. For the local electric measurements, the bias voltage of ±10 V was applied to the sample.

### Magnetic calculation

The magnetic calculations within a spin-polarized frame were carried out with the Vienna *ab* initio Simulation Package (VASP)[53]. The elemental cores and valence electrons were represented by the projector augmented wave (PAW) method. The generalized gradient approximation with the Perdew–Burke–Ernzerhof (GGA-PBE) exchange-correlation functional was employed for all the magnetic calculations[54]. During the calculation process, the energy cutoff and precision energy were set as 700 and $10^{-6}$ eV, respectively. And the force convergence criterion was $10^{-2}$ eV/Å for each atom. Monkhorst-Pack k-points of $2 \times 6 \times 1$ and $1 \times 6 \times 1$ were applied for all the surface calculations of SnSe and $SnSe/SnSe_2$, respectively. For the ferromagnetism of SnSe and $SnSe/SnSe_2$, the initial magnetic moments of Sn atoms were set as +2 $\mu_B$.

### Ferroelectric calculation

The ferroelecrric calculations were performed by VASP with the PBE exchange-correlation function and PAW method. The plane wave cutoff energy was set as 600 eV. The interface was constructed by stacking the $1 \times 3$ supercell of SnSe on $1 \times 2$ supercell of $SnSe_2$. In view of the different atomic structures between SnSe (rectangular lattice with the in-plane lattice constants of ~4.2 and ~4.5 Å) and $SnSe_2$ (hexagonal lattice with the in-plane lattice constants of ~3.8 Å), a distortion was induced at the interface. Monkhorst-Pack k-points of $14 \times 14 \times 1$ and $14 \times 5 \times 1$ were used for monolayer SnSe and $SnSe/SnSe_2$, respectively. The atomic structures were relaxed until the energy and force reach less than $10^{-6}$ eV and $10^{-2}$ eV Å$^{-1}$, respectively. A vacuum layer of ~20 Å was added to minimize the interaction between the periodic images. The van der Waals interaction between SnSe and $SnSe_2$ was corrected by the DFT-D3 method of Grimme. The macroscopic electronic polarization was calculated according to the modern theory of polarization based on the Berry phase.

### Exchange interaction strength calculation

The exchange interaction strength ($J$) was calculated by using the four-state method[55] and then the magnetic states were constructed to be $E_1$ (↑ , ↑) = −468.32202313 eV, $E_2$ (↑ , ↓) = −468.42250830 eV, $E_3$ (↓ , ↑) = −468.42250830 eV, and $E_4$ (↓ , ↓) = −468.32202313 eV, respectively. Accordingly, the $J$ value can be calculated to be -0.2 eV, by using the formula of $J = (E_1 + E_4 − E_2 − E_3)/4S^2$. The spin-polarized calculations were performed by using the VASP based on the PAW method. Electron-ion interactions were described by the standard PAW potentials. The plane wave basis was set as 400 eV to expand the smooth part of wave functions. For the electron-electron exchange and correlation interactions, the functional parametrized PBE (a form of GGA) was used throughout. The van der Waals interaction was described via the DFT-D3BJ method. During the geometry optimizations, the bottom atoms were fixed in the bulk form. And the Brillouin-zone integrations were conducted by using Monkhorst-Pack grids of special points with a separation

of $0.06 \text{ Å}^{-1}$. The convergence criterion for the electronic self-consistent loop was set as $10^{-4}$ eV. The atomic structures were optimized until the residual forces were below $0.03 \text{ eV Å}^{-1}$.

## Dipole moment calculation

The Debye equation, which was defined as $\mu = neR$ (where $\mu$ was the net dipole moment in Debye ($10^{-18}$ esu cm), $n$ was the total number of electrons, $e$ was the charge on an electron ($-4.8 \times 10^{-10}$ esu), and $R$ was the difference in cm (between the "centroids" of positive and negative charges)), was used to calculate the dipole moment of 2D p-doped SnSe[56]. The distribution of electrons on Sn/Se atoms was estimated by the bond-valence method (as described by the following formula of $S_i = \exp[(R_0 - R_i)/B]$, where $R_0$ was an empirical constant (Sn(II)–Se, $R_0 = 2.476$ Å; Sn(IV)–Se, $R_0 = 2.524$ Å), $R_i$ was the length of bond "i" in Å, and $B = 0.37$ Å)[57]. The refined atomic coordinates and selected bond distances were shown in Table S3 and Table S4, respectively.

## Data availability

The data generated in this study are provided as a Source Data file. They are also available from the corresponding authors upon request. Source data are provided with this paper.

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

## Acknowledgements

This work was supported by the National Key R&D Program of China (Nos. 2018YFA0703700, J.H. and 2021YFA1200800, J.S.), National Natural Science Foundation of China (Nos. 91964203, J.H. and 92164103, J.S.), the Beijing National Laboratory for Molecular Sciences (No. BNLMS202001, J.S.), and the Fundamental Research Funds for the Central Universities (No. 2042021kf0029, J.S.). The authors would like to acknowledge the Center for Electron Microscopy at Wuhan University for their substantial supports to JEM-F200 and JEM-NEOARM.

## Author contributions

J.S. and J.H. conceived and supervised the research project. R.D. and Y.W. developed and conducted the PVD growth and transfer of 2D p-doped SnSe with M.C., P.W., H.L., W.F., and L.S.'s assistance. Y.W. constructed the device and electrical transport measurements. R.D., Y.W., M.C., P.W., H.L., W.F., and L.S carried out the OM, XPS, XRD, Raman, AFM, PFM, MFM, and TEM characterizations. R.D. performed the ferroelectric and ferrimagnetic measurements. R.D. did the theoretical calculations. All the authors discussed the results and commented on the manuscript.

## Competing interests

The authors declare no competing interests.
