## [Peer Review File · Nature Communications]

Two-dimensional multiferroic material of metallic p-doped SnSeREVIEWER COMMENTS

Reviewer #1 (Remarks to the Author):

The work by Du and coauthors synthesized SnSe sheets with variable thickness from 9nm to 116nm. They characterized the structure, ferroelectric and magnetic property of the film. They found the SnSe₂ microdomains coexist with SnSe, and proposed that charge transfers from SnSe₂ to SnSe and gives rise to metallic feature and degenerate semiconductors and ferrimagnetic long-range order. However, the results and conclusions are not convincing. See the comments and questions below. I cannot recommend it for publication on Nature Communications.

1. The authors stated that SnSe sheets are stacked on 2D SnSe₂ sheets, and according to Figure 4f the top and bottom SnSe sheets are ferroelectrically stacked with the middle SnSe₂. But why are top and bottom SnSe sheets not antiferroelectrically stacked?
2. The stacking of SnSe sheets themselves is also not clear. SnSe bulk has antiferroelectric phase. If the films synthesized in this work have also antiferroelectric SnSe structure, where does the ferroelectric polarization come from?
3. Even if their assumption of SnSe₂-to-SnSe charge transfer is true in their sample, the amount of SnSe vs SnSe₂ sheets is unclear and the p-doping concentration is also not clear.
4. Figure 3f indicates a semimetal character, and the electron and hole pockets are located at different position along Gamma-X. However, the calculations was done by using DFT with PBE functional which can underestimate band gap and give wrong band edges. As a consequence, it may give the incorrect semimetal character. If this is the case here, the proposed mechanism based on the charge transfer from SnSe₂ to SnSe is doubtful, or at least very minimal.
5. The nature of magnetism in 2D p-doped SnSe was not well understood. What caused the breaking of time reversal symmetry? Even if charge transfer takes place, why shall the film develop magnetic order? Since it's emphasized as 2D p-doped SnSe, what mechanism stabilizes the long-range ferromagnetic order to 300K claimed by the authors?
6. Another puzzle is the supercell they used in their SnSe/SnSe₂ calculation. SnSe has a rectangular lattice with in-plane lattice constants of a=4.2 and b=4.5 Angstroms, and SnSe₂ has a hexagonal lattice with in-plane lattice constants of a=b=3.8 Angstroms. In the Methods section, they mentioned that "The interface was constructed based on the 1×3 supercell of SnSe and 1×2 supercell of SnSe₂." It is impossible for the two supercells of SnSe and SnSe₂ to be reasonably matched without significant distortion.

Reviewer #2 (Remarks to the Author):

This work reports "multiferroic" vdW p-doped SnSe systems. The simultaneous ferroelectricity and magnetism is attributed the local phase segregation of SnSe₂ phase in the SnSe nano plate and the accompanied electron transfer at the phases interface. This work is interesting and should be instructive for the development of the 2D materials and multiferroic research. However these comments should be carefully considered by the authors.

1. It seems to me that the emergence of ferromagnetism is due to the phase segregation of SnSe₂, does this mean that ferromagnetism appears only locally in the SnSe₂ region? How can the authors prove that the material has a long-range magnetic ordering?
2. If the mechanism is what explained in the paper, that is the long range magnetic ordering happened at the phases interface, then the materials will be in composite form, that is locally magnetic ordered nanoregions disperse in the SnSe matrix. Then this is not a single phase 2D multiferroic material.
3. How is the size of separated phase SnSe₂? Strongly suggest to give TEM or other evidence.

Only two different valence state of Sn to prove the existence of two phases of SnSe and SnSe₂ is definitely not enough. To this regard, all the analysis based on the existence of the two phases are arbitrary.

4、 From MT and MH measurement, the materials shows weak ferromagnetism, these measurements going to high temperature is highly recommended to trace the starting temperature for magnetic ordering happening.

5、 How is the supercell setup was doing for calculation of a phase separated system including SnSe and SnSe₂ phases (figure 4f)? I think the interface of the two phases will determine different magnetism of the system.

6、 Highly recommend to measure PFM in box-box form for different voltages applied on the sample to cleanly show the domain switching behavior.

Reviewer #3 (Remarks to the Author):

In the manuscript entitled "Two-dimensional multiferroic material of metallic p-doped SnSe", 2D p-doped SnSe with the property of ferroelectric and ferromagnetic order in a single material. 2D materials based multiferroic heterostructure will help for the coupling. The text and figures satisfy the basic studies for ferrimagnetism and ferroelectricity. The following comments should be considered to improve the manuscript's publication quality and meet the Nature Communication standards.

1. Figure 1 d-f shows the nanosheets were tetragonal and circle. The atomic arrangement of the tetragonal nanosheets was discussed in detail in the manuscript [Page. No-4, Section- the atomic structure of PVD-synthesized 2D SnSe nanosheets]. But the circle nanosheet atomic arrangement is not discussed. The different nucleation leads to the change in the shape of the nanosheet. The nucleation may affect the atomic arrangement, bond length and bond angle. The author can explain the circle nanosheet atomic arrangement in the manuscript in detail.

2. Ferrimagnetism is more associated with permanent magnetism or spontaneous magnetism. The remanence is related to permanent magnetism. But in the report (figure 4b), the remanence seems low. This suggests the low permanent magnetism in the prepared sample. The author should justify the above-stated point with the proper explanation.

3. The remanence, coercivity and saturation magnetism are the key parameters. The detailed contrast is not in the manuscript. A discussion in detail is required.

4. The magnitude of the dipole moment calculation using the bond-valence method and the Debye equation gives a better understanding of the material toward ferroelectricity and ferrimagnetism.

5. The author should report the Raman spectrum till 220 Raman shift in figure 1b. The predominant A_{1g} band of SnSe₂ were observed near 200 Raman shift. The authors can confirm the microdomain observation of SnSe₂.

Reviewer #1 (Remarks to the Author):

The work by Du and coauthors synthesized SnSe sheets with variable thickness from 9 nm to 116 nm. They characterized the structure, ferroelectric and magnetic property of the film. They found the SnSe₂ microdomains coexist with SnSe, and proposed that charge transfers from SnSe₂ to SnSe and gives rise to metallic feature and degenerate semiconductors and ferrimagnetic long-range order. However, the results and conclusions are not convincing. See the comments and questions below. I cannot recommend it for publication on Nature Communications.

Our response:

We are very thankful for the reviewer's kind suggestion and constructive comments. The detailed discussion regarding the origin of ferroelectricity and ferrimagnetism in 2D p-doped SnSe is added, additional theoretical and experimental results are also provided, according to the reviewer's comments. In addition, these issues raised by the reviewer are considered very carefully and addressed point-by-point as follows.

1. The authors stated that SnSe sheets are stacked on 2D SnSe₂ sheets, and according to Figure 4f the top and bottom SnSe sheets are ferroelectrically stacked with the middle SnSe₂. But why are top and bottom SnSe sheets not antiferroelectrically stacked?

Our response:

We are very thankful for the reviewer's constructive comment.

We have performed additional DFT calculations regarding the following two cases: ferroelectric stack with the middle SnSe₂ and antiferroelectric stack with the middle SnSe₂, as shown in **Figure R1**. We find that the energy of ferroelectrically stacked case is ~3.23 meV lower than that of antiferroelectrically stacked configuration. Therefore, the top and bottom SnSe nanosheets are ferroelectrically stacked with the middle SnSe₂. We have provided the calculated results in **Fig. S21** (or **Figure R1**) and added some discussion in **page 11** by “...To determine the stacked configuration of top and bottom SnSe nanosheets, DFT calculations were performed with the results shown in **Fig. S21**. Interestingly, the energy of ferroelectric stack is ~3.23 meV lower than that of the antiferroelectrically stacked structure, and thus the ferroelectrically staked configuration is established...”.

Figure R1 (or Fig. S21) | The energies of ferroelectric and antiferroelectric stacking of SnSe/SnSe₂. The much smaller energy of ferroelectric stack suggests its relatively stable structure.

2. The stacking of SnSe sheets themselves is also not clear. SnSe bulk has antiferroelectric phase. If the films synthesized in this work have also antiferroelectric SnSe structure, where does the ferroelectric polarization come

from?

Our response:

We are very thankful for the reviewer's constructive comment.

We agree with the reviewer that the antiferroelectric phase is relatively stable for bulk SnSe, we have also calculated the corresponding energy at ferroelectric (−34.5805 eV) and antiferroelectric (−34.6414 eV) stack, respectively. Nevertheless, with decreasing the layer thickness, the energy difference between ferroelectric and antiferroelectric stack is reduced accordingly. For the bilayer SnSe, the energy difference (∼30.40 meV) between ferroelectric (−34.0225 eV) and antiferroelectric (−34.0529 eV) stack is reduced to the half of bulk counterpart (∼60.90 meV). Particularly, after intercalating the SnSe₂ nanosheet, the energy of ferroelectric stack is ∼3.23 meV lower than that of the antiferroelectrically stacked structure, and thus the ferroelectrically stacked configuration is established

The ferroelectric polarization in 2D p-doped SnSe possibly originates from the strain/lattice distortion and subsequent off-center displacement of Sn, due to the phase segregation of SnSe₂ microdomains. The different chemical environment of Sn at the interfaces leads to the formation of uncompensated charge transfer and dipole (*ACS Nano* **15**, 9229–9237 (2021); *Natl. Sci. Rev.* **6**, 653–668 (2019)). Meanwhile, the corresponding ionic-potential anharmonicity in 2D p-doped SnSe may also contribute to the emergence of ferroelectric polarization (*Phys. Rev. Lett.* **117**, 097601 (2016); *Nano Lett.* **16**, 3236–3241 (2016)). We have added some discussion in **page 12** by “...*The ferroelectric polarization in 2D p-doped SnSe possibly originates from the strain/lattice distortion and subsequent off-center displacement of Sn, due to the phase segregation of SnSe₂ microdomains. The different chemical environment of Sn at the interfaces leads to the formation of uncompensated charge transfer and dipole^{41,42}. In addition, the corresponding ionic-potential anharmonicity in 2D p-doped SnSe may also contribute to the emergence of ferroelectric polarization^{29,43} ...*”.

We have cited new references in **page 12** (Refs. 41–43).

3. Even if their assumption of SnSe₂-to-SnSe charge transfer is true in their sample, the amount of SnSe vs SnSe₂ sheets is unclear and the p-doping concentration is also not clear.

Our response:

We are very thankful for the reviewer's constructive comment.

According to the reviewer's suggestion, we have performed TEM measurements to identify the domain size and amount of SnSe₂ microdomains, with the results shown in **Fig. S13,14** (or **Figure R2** and **Figure R3**). Interestingly, hexagonal SnSe₂ microdomains are obviously observed with the domain size changing from ∼23 to ∼105 nm and average coverage of ∼15.6%. We have provided the TEM results in **Fig. S13,14** (or **Figure R2** and **Figure R3**) and added some discussion in **page 7** by “...*To further confirm the formation of SnSe₂ microdomains in SnSe, TEM measurements were then performed and hexagonal SnSe₂ microdomains with the domain size changing from ∼23 to ∼105 nm and average coverage of ∼15.6% were obviously observed (Fig. S13,14)....*”.

In addition, we have also performed Hall measurements to extract the carrier concentration of 2D p-doped SnSe (∼8.31 × 10¹⁹ cm⁻³ at 150 K and ∼5.70 × 10¹⁹ cm⁻³ at 2 K), with the results shown in **Fig. S10** (or **Figure R4**). We have added this discussion in **page 7** by “...*In addition, the carrier concentration of 2D p-doped SnSe is extracted to be ∼8.31 × 10¹⁹ cm⁻³ at 150 K and ∼5.70 × 10¹⁹ cm⁻³ at 2 K, by performing the Hall measurements (Fig. S10)....*”.

Figure R2 (or Fig. S13) | The determination of SnSe₂ microdomains by TEM. (a) Low-magnification TEM image of a p-doped SnSe nanosheet, showing the formation of SnSe₂ microdomains. (b) Corresponding SAED pattern captured from the SnSe₂ region. (c) Atomic-resolution TEM image obtained from the SnSe₂ area.

Figure R3 (or Fig. S14) | The coverage determination of SnSe₂ microdomains. (a–c) Low-magnification TEM images captured from different areas of p-doped SnSe. (d) Statistical distribution of the coverage of SnSe₂.

Figure R4 (or Fig. S10) | Temperature-dependent carrier concentration of 2D p-doped SnSe.

4. Figure 3f indicates a semimetal character, and the electron and hole pockets are located at different position along Gamma-X. However, the calculations were done by using DFT with PBE functional which can underestimate band gap and give wrong band edges. As a consequence, it may give the incorrect semimetal character. If this is the case here, the proposed mechanism based on the charge transfer from SnSe₂ to SnSe is doubtful, or at least very minimal.

Our response:

We are very thankful for the reviewer's constructive comment.

According to the reviewer's suggestion, we have performed the band structure calculation by using DFT with HSE function in **Fig. S15** (or **Figure R5**). The metallic character is reconfirmed in 2D p-doped SnSe. Moreover, according to the transfer characteristic curves in **Fig. 3b** and the temperature-dependent longitudinal resistances in **Fig. 3c**, the metallic feature of 2D p-doped SnSe is clearly observed. In addition, Wang *et al.* also performed the ARPES measurements and DFT calculations to explore the band structure of p-doped SnSe, and the metallic behavior was also obtained (*Nat. Commun.* **9**, 47 (2018)). Based on the theoretical and experimental results, we can confirm the metallic property of 2D p-doped SnSe. We have provided the band structures obtained by using DFT with HSE function in **Fig. S15** (or **Figure R5**) and added some discussion in **page 8** by "...*The band structures of SnSe/SnSe₂ calculated with different methods are presented in Fig. 3f and Fig. S15, respectively, and the Fermi level is crossed by several bands, indicative of its metallic feature, echoing well with the experimental findings....*".

Figure R5 (or Fig. S15) | Calculated the band structure of SnSe/SnSe₂ by using DFT with HSE function. The Fermi level is crossed by a band (indicated by the red circle), indicative of its metallic feature.

5. The nature of magnetism in 2D p-doped SnSe was not well understood. What caused the breaking of time reversal symmetry? Even if charge transfer takes place, why shall the film develop magnetic order? Since it's emphasized as 2D p-doped SnSe, what mechanism stabilizes the long-range ferromagnetic order to 300K claimed by the authors?

Our response:

We are very thankful for the reviewer's constructive comment.

The physical origin of magnetism in 2D p-doped SnSe can be understood by the Stoner mechanism. In detail, within the band-picture model, the spontaneous ferromagnetic order is appeared when the exchange interaction is larger than the loss of kinetic energy, that is, when it is satisfying the "Stoner Criterion": $D_{(E_f)}J > 1$, where $D_{(E_f)}$ is the DOS at the Fermi energy and J is the exchange interaction strength (*Phys. Rev. Lett.* **114**, 236602 (2015)). And thus, we have calculated the DOS at the Fermi energy ($D_{(E_f)} = 9.35$ states/eV) and the exchange interaction strength ($J = 0.20$ eV) of 2D p-doped SnSe. Obviously, the "Stoner Criterion" is well satisfied ($D_{(E_f)}J = 1.87 > 1$) and then the ferromagnetism is formed in 2D p-doped SnSe. Notably, the relatively high $D_{(E_f)}$ in 2D p-doped SnSe may lead to the exchange-field splitting near the VBM and contributes to the formation of ferromagnetism. Additionally, the phase segregation of SnSe₂ microdomains introduces the strain/lattice distortion, which possibly modulates the Berry curvature dipole and results in the appearance of orbital magnetism (*Chin. Phys. Lett.* **38**, 017301 (2021)). We have added some discussion

in **page 10** by “...*The physical origin of magnetism in 2D p-doped SnSe can be understood by the Stoner mechanism. In detail, the DOS at the Fermi energy ($D_{(E_f)} = 9.35$ states/eV) and the exchange interaction strength ($J = 0.20$ eV) are calculated for 2D p-doped SnSe, and the “Stoner Criterion” ($D_{(E_f)}J = 1.87 > 1$) is found to be well-satisfied, suggesting that the exchange interaction is larger than the loss of kinetic energy and thus induces the formation of ferromagnetism³³. Notably, the relatively high $D_{(E_f)}$ in 2D p-doped SnSe may lead to the exchange-field splitting near the VBM and contributes to the formation of ferromagnetism. In addition, the phase segregation of SnSe₂ microdomains introduces the strain/lattice distortion, which possibly modulates the Berry curvature dipole and results in the appearance of orbital magnetism³⁴....”.*

In consideration of the Stoner ferromagnet feature of 2D p-doped SnSe, the time reversal symmetry (TRS) is possibly broken by the spontaneous magnetic field, where the self-consistent exchange-field leads to the net magnetization. Furthermore, the anomalous asymmetric Jahn-Teller distortion may be occurred at the interface, which should induce the charge/orbital orders. The spatial electron-hole separation and the spontaneous symmetry breaking result in the emergence of ferromagnetism in 2D p-doped SnSe (*Phys. Rev. Lett.* **120**, 147601 (2018)).

The high temperature long-range ferromagnetic order in 2D p-doped SnSe can be explained by the kinetic exchange mechanism (*Phys. Rev. Lett.* **100**, 117204 (2008)). In detail, during the charge transfer process, the majority spin is fully occupied, where the minority spin is partially occupied. And then, the parallel spin alignment allows for the hopping between the adjacent states, which reduces the kinetic energy and stabilizes the long-range ferromagnetic order at high temperature (~300 K). The similar mechanism is also proposed in monolayer nonmagnetic p-doped GaSe (*Phys. Rev. Lett.* **114**, 236602 (2015)). Additionally, in view of the large domain size and high coverage of SnSe₂ microdomains (**Fig. S13,14**), the abundant phase interfaces are evolved, which enhance the exchange interactions of magnetic moments (mainly locate at the interfaces) and contribute to the emergence of long-range ferromagnetic order in 2D p-doped SnSe. We have added some discussion in **page 9** by “...*The high temperature long-range ferromagnetic order in 2D p-doped SnSe can be explained by the kinetic exchange mechanism³². In detail, during the charge transfer process, the parallel spin alignment contributes to the hopping between the adjacent states, which reduces the kinetic energy and stabilizes the long-range ferromagnetic order at high temperature (~300 K). Besides, in view of the abundant interfaces between SnSe and SnSe₂, the exchange interactions of magnetic moments at the interfaces are enhanced and contribute to the emergence of long-range ferromagnetic order in 2D p-doped SnSe....”.*

We have cited new references in **page 9** and **page 10** (Ref. 32–34).

6. Another puzzle is the supercell they used in their SnSe/SnSe₂ calculation. SnSe has a rectangular lattice with in-plane lattice constants of $a=4.2$ and $b=4.5$ Angstroms, and SnSe₂ has a hexagonal lattice with in-plane lattice constants of $a=b=3.8$ Angstroms. In the Methods section, they mentioned that "The interface was constructed based on the 1×3 supercell of SnSe and 1×2 supercell of SnSe₂." It is impossible for the two supercells of SnSe and SnSe₂ to be reasonably matched without significant distortion.

Our response:

We are very thankful for the reviewer’s constructive comment.

We agree with the reviewer that the atomic structures of SnSe₂ and SnSe are incompatible, the lattice constant mismatch between SnSe₂ and SnSe is calculated to be ~9.5% and a significant distortion should be induced at the interfaces. Accordingly, a large strain will be introduced in SnSe when the supercells of SnSe and SnSe₂ are both set

as 1×1 . In order to fully eliminate this strain in SnSe, the supercell of SnSe and SnSe₂ along a direction should be set as 10 and 11, respectively. However, such a large supercell requires enormous computational resources. And thus, in our manuscript, the interface is constructed based on the 1×3 supercell of SnSe and 1×2 supercell of SnSe₂, which is the largest supercell we can simulate. In addition, these two materials with distinct lattice constants and crystal symmetries can be connected seamlessly by changing the bond length and rotation angle, as has been confirmed by STM (*ACS Appl. Mater. Interfaces* **10**, 12831–12838 (2018)). We have added some discussion in **page 14** by “...*In view of the different atomic structures between SnSe (rectangular lattice with the in-plane lattice constants of ~ 4.2 and ~ 4.5 Å) and SnSe₂ (hexagonal lattice with the in-plane lattice constants of ~ 3.8 Å), a distortion may be induced at the interface. Nevertheless, these two materials with distinct lattice constants and crystal symmetries can be connected seamlessly by changing the bond length and rotation angle⁴⁸*”.

We have cited new references in **page 14** (Ref. 48).

Reviewer #2 (Remarks to the Author):

This work reports “multiferroic” vdW p-doped SnSe systems. The simultaneous ferroelectricity and magnetism is attributed the local phase segregation of SnSe₂ phase in the SnSe nanoplate and the accompanied electron transfer at the phases interface. This work is interesting and should be instructive for the development of the 2D materials and multiferroic research. However, these comments should be carefully considered by the authors.

Our response:

We are very grateful for the reviewer’s positive evaluation toward the significance of our manuscript. We also appreciate the reviewer’s very kind suggestion and constructive comments. These issues raised by the reviewer are considered very carefully and addressed point-by-point as follows.

1. It seems to me that the emergence of ferromagnetism is due to the phase segregation of SnSe₂, does this mean that ferromagnetism appears only locally in the SnSe₂ region? How can the authors prove that the material has a long-range magnetic ordering?

Our response:

We are very thankful for the reviewer’s constructive comment.

We agree with the reviewer that the phase segregation of SnSe₂ is possibly one of the reasons for the emergence of ferromagnetism. Even so, we think that the ferromagnetism is not appeared only locally in the SnSe₂ region. The phase segregation of SnSe₂ microdomains introduces the strain/lattice distortion, which possibly modulates the Berry curvature dipole and results in the appearance of orbital magnetism (*Chin. Phys. Lett.* **38**, 017301 (2021)). Notably, the large domain size and high coverage of SnSe₂ microdomains are obviously observed (**Fig. S13,14**) and then the abundant interfaces are evolved accordingly, which enhance the exchange interactions of magnetic moments and contribute to the emergence of long-range ferromagnetic order in 2D p-doped SnSe. Besides, the physical origin of ferromagnetism in 2D p-doped SnSe can also be explained by the Stoner mechanism. In detail, within the band-picture model, the spontaneous ferromagnetic order is appeared when the exchange interaction is larger than the loss of kinetic energy, that is, when it is satisfying the “Stoner Criterion”: $D_{(\text{EF})}J > 1$, where $D_{(\text{EF})}$ is the DOS at the Fermi energy and J is the exchange interaction strength (*Phys. Rev. Lett.* **114**, 236602 (2015)). And thus, we have calculated the DOS at the Fermi energy ($D_{(\text{EF})} = 9.35$ states/eV) and the exchange interaction strength ($J = 0.20$ eV) of 2D p-doped SnSe. Obviously, the “Stoner Criterion” is well satisfied ($D_{(\text{EF})}J = 1.87 > 1$) and the ferromagnetism is formed

in 2D p-doped SnSe.

According to the SQUID results in **Fig. 4a–d**, the long-range ferromagnetic order is confirmed in 2D p-doped SnSe, and which can be explained by the kinetic exchange mechanism (*Phys. Rev. Lett.* **100**, 117204 (2008)). In detail, during the charge transfer process, the majority spin is fully occupied, where the minority spin is partially occupied. And then, the parallel spin alignment allows for the hopping between the adjacent states, which reduces the kinetic energy and stabilizes the long-range ferromagnetic order at high temperature (~300 K). The similar mechanism is also proposed in monolayer nonmagnetic p-doped GaSe (*Phys. Rev. Lett.* **114**, 236602 (2015)). We have added some discussion in **page 9** by “...*The high temperature long-range ferromagnetic order in 2D p-doped SnSe can be explained by the kinetic exchange mechanism³². In detail, during the charge transfer process, the parallel spin alignment contributes to the hopping between the adjacent states, which reduces the kinetic energy and stabilizes the long-range ferromagnetic order at high temperature (~300 K). Besides, in view of the abundant interfaces between SnSe and SnSe₂, the exchange interactions of magnetic moments at the interfaces are enhanced and contribute to the emergence of long-range ferromagnetic order in 2D p-doped SnSe....*”.

2. If the mechanism is what explained in the paper, that is the long-range magnetic ordering happened at the phases interface, then the materials will be in composite form, that is locally magnetic ordered nanoregions disperse in the SnSe matrix. Then this is not a single phase 2D multiferroic material.

Our response:

We are very thankful for the reviewer’s constructive comment.

We agree with the reviewer that the phase segregation of SnSe₂ is possibly one of the reasons for the emergence of ferromagnetism. Even so, we think that the long-range magnetic order is not happened only at the nanoregions. The phase segregation of SnSe₂ microdomains introduces the strain/lattice distortion, which possibly modulates the Berry curvature dipole and results in the appearance of orbital magnetism (*Chin. Phys. Lett.* **38**, 017301 (2021)). Notably, the large domain size and high coverage of SnSe₂ microdomains are obviously observed (**Fig. S13,14**) and then the abundant interfaces are evolved accordingly, which enhance the exchange interactions of magnetic moments and contribute to the emergence of long-range ferromagnetic order in 2D p-doped SnSe.

According to the SQUID results in **Fig. 4a–d**, the long-range ferromagnetic order is confirmed in 2D p-doped SnSe, and which can be explained by the kinetic exchange mechanism (*Phys. Rev. Lett.* **100**, 117204 (2008)). In detail, during the charge transfer process, the majority spin is fully occupied, where the minority spin is partially occupied. And then, the parallel spin alignment allows for the hopping between the adjacent states, which reduces the kinetic energy and stabilizes the long-range ferromagnetic order at high temperature (~300 K). The similar mechanism is also proposed in monolayer nonmagnetic p-doped GaSe (*Phys. Rev. Lett.* **114**, 236602 (2015)). We have added some discussion in **page 9** by “...*The high temperature long-range ferromagnetic order in 2D p-doped SnSe can be explained by the kinetic exchange mechanism³². In detail, during the charge transfer process, the parallel spin alignment contributes to the hopping between the adjacent states, which reduces the kinetic energy and stabilizes the long-range ferromagnetic order at high temperature (~300 K). Besides, in view of the abundant interfaces between SnSe and SnSe₂, the exchange interactions of magnetic moments at the interfaces are enhanced and contribute to the emergence of long-range ferromagnetic order in 2D p-doped SnSe....*”.

Based on the abovementioned theoretical and experimental results, we think that the multiferroic feature is appeared in the single phase of 2D p-doped SnSe.

3. How is the size of separated phase SnSe₂? Strongly suggest to give TEM or other evidence. Only two different valence state of Sn to prove the existence of two phases of SnSe and SnSe₂ is definitely not enough. To this regard, all the analysis based on the existence of the two phases are arbitrary.

Our response:

We are very thankful for the reviewer's constructive comment.

According to the reviewer's suggestion, we have performed TEM measurements to identify the domain size and distribution of SnSe₂ microdomains, with the results shown in **Fig. S13,14** (or **Figure R6** and **Figure R7**). Interestingly, hexagonal SnSe₂ microdomains are obviously observed with the domain size changing from ~23 to ~105 nm and average coverage of ~15.6%. We have provided the TEM results in **Fig. S13,14** (or **Figure R6** and **Figure R7**) and added some discussion in **page 8** by "...To further confirm the formation of SnSe₂ microdomains in SnSe, TEM measurements were then performed and hexagonal SnSe₂ microdomains with the domain size changing from ~23 to ~105 nm and average coverage of ~15.6% were obviously observed (**Fig. S13,14**)....".

Figure R6 (or Fig. S13) | The determination of SnSe₂ microdomains by TEM. (a) Low-magnification TEM image of a p-doped SnSe nanosheet, showing the formation of SnSe₂ microdomains. (b) Corresponding SAED pattern captured from the SnSe₂ region. (c) Atomic-resolution TEM image obtained from the SnSe₂ area.

Figure R7 (or Fig. S14) | The coverage determination of SnSe₂ microdomains. (a–c) Low-magnification TEM images captured from different areas of p-doped SnSe. (d) Statistical distribution of the coverage of SnSe₂.

4. From MT and MH measurement, the materials show weak ferromagnetism, these measurements going to high

temperature is highly recommended to trace the starting temperature for magnetic ordering happening.

Our response:

We are very thankful for the reviewer's constructive comment.

According to the reviewer's suggestion, we have performed the temperature-dependent magnetic moment measurements of 2D p-doped SnSe from ~2 to ~700 K and the ferromagnetic order is disappeared at ~665 K. We have provided the results in **Fig. S18** (or **Figure R8**) and added some discussion in **page 9** by "...*To further trace the magnetic order evolution of 2D p-doped SnSe, the temperature-dependent magnetic moment measurements (from ~2 to ~700 K) were performed, and the ferromagnetic order was disappeared at ~665 K, reconfirming the high temperature ferrimagnetic property of 2D p-doped SnSe (Fig. S18)....*".

Figure R8 (or Fig. S18) | Temperature-dependent magnetic moment of 2D p-doped SnSe nanosheets with a vertical and parallel magnetic field at 100 Oe. The ferromagnetic order is disappeared at ~665 K, reconfirming the high temperature ferrimagnetic property of 2D p-doped SnSe.

In addition, we have also provided the magnetic hysteresis loops of 2D p-doped SnSe nanosheets at the temperatures of ~5, ~300, and ~400 K under the parallel and vertical magnetic fields, and the long-range ferrimagnetic order is convinced, as shown in **Fig. 4b,d** (or **Figure R9**). We have added some discussion in **page 9** by "...*The magnetic hysteresis loops of 2D p-doped SnSe nanosheets are clearly achieved at the temperatures of ~5, ~300, and ~400 K under the parallel and vertical magnetic fields (Fig. 4b,d), confirming the long-range ferrimagnetic order....*".

Figure R9 (or Fig. 4b,d) | Magnetic hysteresis loops of 2D p-doped SnSe at ~5, ~300, and ~400 K with the magnetic field vertical and parallel to c-axis, respectively.

5. How is the supercell setup was doing for calculation of a phase separated system including SnSe and SnSe₂ phases (figure 4f)? I think the interface of the two phases will determine different magnetism of the system.

Our response:

We are very thankful for the reviewer's constructive comment.

The atomic structures of SnSe₂ and SnSe are incompatible, the lattice constant mismatch between SnSe₂ and SnSe is calculated to be ~9.5% and a significant distortion should be induced at the interfaces. Accordingly, a large strain will be introduced in SnSe when the supercells of SnSe and SnSe₂ are both set as 1 × 1. In order to fully eliminate this strain in SnSe, the supercell of SnSe and SnSe₂ along *a* direction should be set as 10 and 11, respectively. However, such a large supercell requires enormous computational resources. And thus, in our manuscript, the interface is constructed based on the 1 × 3 supercell of SnSe and 1 × 2 supercell of SnSe₂, which is the largest supercell we can simulate. In addition, these two materials with distinct lattice constants and crystal symmetries can be connected seamlessly by changing the bond length and rotation angle, as has been confirmed by STM (*ACS Appl. Mater. Interfaces* **10**, 12831–12838 (2018)). We have added some discussion in **page 14** by “...*In view of the different atomic structures between SnSe (rectangular lattice with the in-plane lattice constants of ~4.2 and ~4.5 Å) and SnSe₂ (hexagonal lattice with the in-plane lattice constants of ~3.8 Å), a distortion may be induced at the interface. Nevertheless, these two materials with distinct lattice constants and crystal symmetries can be connected seamlessly by changing the bond length and rotation angle⁴⁸*”.

We have cited new references in **page 14** (Ref. 48).

6. Highly recommend to measure PFM in box-box form for different voltages applied on the sample to cleanly show the domain switching behavior.

Our response:

We are very thankful for the reviewer’s constructive comment.

According to the reviewer’s suggestion, we have performed the ferroelectric polarization switching measurements of 2D p-doped SnSe with the opposite bias (±10 V), and the reversal of phase contrast is obviously observed, highly indicatively of the robust and intrinsic ferroelectric feature in 2D p-doped SnSe, as well as its switching behavior. We have provided the ferroelectric polarization switching results in **Fig. 5b,c** (or **Figure R10**) and added some discussion in **page 10** and **page 11** by “...*Figure 5b,c reveal the PFM phase and amplitude images captured from 2D p-doped SnSe after writing a box-in-box pattern with the opposite bias (±10 V). Interestingly, the PFM phase image (Fig. 5b) reveal bright and dark regions with a nearly 180° phase contrast, corresponding to the upwards and downwards polarization states, respectively. Meanwhile, an enhanced PFM amplitude signal is obtained in the electrically poled region (Fig. 5c, white dashed square), the clear and well-defined domain wall is observed on the boundary. Such results indicate the robust and intrinsic ferroelectric feature in 2D p-doped SnSe, as well as its switching behavior....*”.

Figure R10 (or Fig. 5b,c) | Phase and amplitude images of 2D p-doped SnSe after poling with ±10 V, demonstrating stable polarization states.

Reviewer #3 (Remarks to the Author):

In the manuscript entitled “Two-dimensional multiferroic material of metallic p-doped SnSe”, 2D p-doped SnSe with the property of ferroelectric and ferromagnetic order in a single material. 2D materials based multiferroic heterostructure will help for the coupling. The text and figures satisfy the basic studies for ferrimagnetism and ferroelectricity. The following comments should be considered to improve the manuscript's publication quality and meet the Nature Communication standards.

Our response:

We are very grateful for the reviewer's positive evaluation toward the significance of our manuscript. We also appreciate the reviewer's very kind suggestion and constructive comments. These issues raised by the reviewer are considered very carefully and addressed point-by-point as follows.

1. Figure 1d-f shows the nanosheets were tetragonal and circle. The atomic arrangement of the tetragonal nanosheets was discussed in detail in the manuscript [Page. No-4, Section- the atomic structure of PVD-synthesized 2D SnSe nanosheets]. But the circle nanosheet atomic arrangement is not discussed. The different nucleation leads to the change in the shape of the nanosheet. The nucleation may affect the atomic arrangement, bond length and bond angle. The author can explain the circle nanosheet atomic arrangement in the manuscript in detail.

Our response:

We are very thankful for the reviewer's constructive comment.

We agree with the reviewer that the nucleation leads to the change in the shape of SnSe and its atomic arrangement. According to the reviewer's suggestion, we have provided the TEM results of circle SnSe nanosheets in **Fig. S7** (or **Figure R11**), in which the same atomic arrangement with tetragonal SnSe is obviously observed. Nevertheless, the relatively high density of defect is obtained, and a similar result is also demonstrated in PVD-synthesized circle SnS (*Nat. Commun.* **11**, 2428 (2020)). We have provided the TEM results of circle SnSe in **Fig. S7** (or **Figure R11**) and added some discussion in **page 6** by “...*The atomic structure of circle SnSe is also detected and shown in Fig. S7, where the same atomic arrangement with tetragonal SnSe is clearly obtained. Nevertheless, the relatively high density of defect is observed due to its low crystalline quality, and a similar result is also demonstrated in PVD-synthesized 2D circle SnS²¹....*”.

Figure R11 (or Fig. S7) | TEM characterization of a circle SnSe nanosheet. (a) Low-magnification TEM image of a circle SnSe nanosheet. (b) Corresponding atomic-resolution TEM image.

2. Ferrimagnetism is more associated with permanent magnetism or spontaneous magnetism. The remanence is related to permanent magnetism. But in the report (figure 4b), the remanence seems low. This suggests the low

permanent magnetism in the prepared sample. The author should justify the above-stated point with the proper explanation.

Our response:

We are very thankful for the reviewer’s constructive comment.

We agree with the reviewer that the ferrimagnetism is associated with the permanent or spontaneous magnetism. For 2D p-doped SnSe, the relatively low remanence suggests the weak permanent magnetism. In view of the 2D character and the charge transfer between SnSe and SnSe₂ microdomains, the low remanence is accessible. The relatively weak magnetism in 2D p-doped SnSe can be explained by the Stoner mechanism (*Phys. Rev. Lett.* **114**, 236602 (2015)), where the exchange interaction is larger than the loss of kinetic energy. In addition, the phase segregation of SnSe₂ microdomains is possibly another reason for interpreting the emergence of ferrimagnetism, as has been demonstrated in the transition metal atoms doped monolayer SnSe (*J. Phys. D: Appl. Phys.* **51** 245004 (2018)). We have added this discussion in **page 9** by “...*Notably, a relatively low remanence is obtained in 2D p-doped SnSe, suggesting the weak permanent magnetism and ferrimagnetism....*”.

3. The remanence, coercivity and saturation magnetism are the key parameters. The detailed contrast is not in the manuscript. A discussion in detail is required.

Our response:

We are very thankful for the reviewer’s constructive comment.

According to the reviewer’s kind suggestion, we have provided a table (**Table S1**) including the remanence, coercivity, and saturation magnetism of 2D p-doped SnSe, and added some discussion in **page 9** by “...*The corresponding remanence, coercivity, and saturation magnetism are also extracted and shown in Table S1....*”.

Table R1 (or Table S1). The remanence, coercivity, and saturation magnetism of 2D p-doped SnSe

Temperature (K)	In plane			Out of plane		
	Remanence (μemu)	Coercivity (T)	Saturation magnetism (μemu)	Remanence (μemu)	Coercivity (T)	Saturation magnetism (μemu)
300	2.2	0.009	123.4	24.0	0.02	134.1
5	107.5	0.018	528.7	3.6	0.02	113.2

4. The magnitude of the dipole moment calculation using the bond-valence method and the Debye equation gives a better understanding of the material toward ferroelectricity and ferrimagnetism.

Our response:

We are very thankful for the reviewer’s constructive comment.

According to the reviewer’s suggestion, we have calculated the dipole moments of 2D p-doped SnSe by using the bond-valence method and Debye equation as follows: $\mu = q \times l$, $P = \mu/V$, where μ is the dipole moment, q is the negative charge, l is the actual distance between the positive and negative charges, P is the magnitude of the dipole moment, and V is the unit cell volume. Interestingly, after intercalating the SnSe₂ layer, the dipole moment (~0.0001 Debye) is reduced comparing with that of intrinsic SnSe (~0.0510 Debye), indicative of the weak ferroelectricity in 2D p-

doped SnSe, consistent with the experimental results. We have provided the calculated results in **Fig. S24** (or **Figure R12**) and added some discussion in **page 12** by “...Meanwhile, the dipole moments of 2D p-doped SnSe are also calculated by using the bond-valence method and Debye equation (**Fig. S24**) and the reduced dipole moment value (~ 0.0001 Debye) comparing with that of intrinsic SnSe (~ 0.0510 Debye) suggestive of the weak ferroelectricity in 2D p-doped SnSe, consistent with the experimental results....”.

In addition, the magnetic moments of 2D p-doped SnSe ($\sim 0.7881 \mu_B$) and intrinsic SnSe ($\sim 0.5175 \mu_B$) are also calculated and the much higher value indicates the emergence of ferrimagnetism in PVD-synthesized 2D p-doped SnSe nanosheets. We have also added this discussion in **page 10** by “...The projected spin-polarized DOS in **Fig. 4f** reveals that the states around Fermi level are determined by Se atoms, and the asymmetric DOS of two spin channels results in a total magnetic moment of $\sim 0.7881 \mu_B$, much higher than that of intrinsic SnSe ($\sim 0.5175 \mu_B$), indicatively of the emergence of ferrimagnetism in PVD-synthesized 2D p-doped SnSe nanosheets....”.

Figure R12 (or Fig. S24) | Atomic structures of trilayer SnSe and SnSe/SnSe₂. The dipole moments of 2D p-doped SnSe are calculated by using the bond-valence method and Debye equation. The small dipole moment value comparing with that of intrinsic SnSe indicates the weak ferroelectricity in 2D p-doped SnSe, consistent with the experimental results.

5. The author should report the Raman spectrum till 220 Raman shift in figure 1b. The predominant A_{1g} band of SnSe₂ were observed near 200 Raman shift. The authors can confirm the microdomain observation of SnSe₂.

Our response:

We are very thankful for the reviewer’s kind suggestion.

We have revised the Raman spectrum of 2D p-doped SnSe in **Fig. 1b** to show the A_{1g} band of SnSe₂. Notably, the phase segregation of SnSe₂ microdomains is occurred inside of SnSe, and Raman is a surface determination technology, therefore, the characteristic peak of SnSe₂ is not fully displayed. In order to identify the formation of SnSe₂ microdomains, the samples are treated by Ar plasma and characterized by Raman. Interestingly, the A_{1g} band of SnSe₂ are obviously observed at $\sim 180 \text{ cm}^{-1}$ (**Fig. 3e**). Furthermore, the Raman characterizations of transfer flipped SnSe nanosheets are also performed, and the characteristic peaks of SnSe₂ are obtained, reconfirming the formation of SnSe₂ microdomains (**Fig. S12**).

In addition, we have also performed TEM measurements to identify the domain size and distribution of SnSe₂ microdomains, with the results shown in **Fig. S13,14** (or **Figure R13** and **Figure R14**). Interestingly, hexagonal SnSe₂ microdomains are obviously observed with the domain size changing from ~ 23 to ~ 105 nm and average coverage of $\sim 15.6\%$. We have provided the TEM results in **Fig. S13,14** (or **Figure R13** and **Figure R14**) and added some discussion in **page 8** by “...To further confirm the evolution of SnSe₂ microdomains in SnSe, TEM measurements

were then performed and hexagonal SnSe_2 microdomains with the domain size changing from ~ 23 to ~ 100 nm and average coverage of $\sim 15.6\%$ were obviously observed (Fig. S13,14)....”.

Figure R13 (or Fig. S13) | The determination of SnSe_2 microdomains by TEM. (a) Low-magnification TEM image of a p-doped SnSe nanosheet, showing the formation of SnSe_2 microdomains. (b) Corresponding SAED pattern captured from the SnSe_2 region. (c) Atomic-resolution TEM image obtained from the SnSe_2 area.

Figure R14 (or Fig. S14) | The coverage determination of SnSe_2 microdomains. (a–c) Low-magnification TEM images captured from different areas of p-doped SnSe. (d) Statistical distribution of the coverage of SnSe_2 .

Reviewers' comments:

Reviewer #1 (Remarks to the Author):

The authors did not address several questions raised in the previous comments. I cannot recommend it for publication on Nature Communications.

1. Regarding the authors' reply to question #1: For the specific two geometries of antiferroelectric and ferroelectric phase, the antiferroelectric phase is only 3.23 meV higher than ferroelectric phase. Moreover, as the authors mentioned, the supercell was significantly stretched in their calculations. How can the conclusion be safely drawn?

2. Regarding the authors' reply to question #2: The authors did not answer my question - "The stacking of SnSe sheets themselves is also not clear." What's the stacking geometry in experiment? As the authors stated in the manuscript, the thickness varies from 3.4 nm to 114 nm. When the SnSe/SnSe₂ layers are formed, how many SnSe layers and SnSe₂ layers are in the final structure? and, what's the stacking geometry of SnSe layers? These questions were raised mainly regarding the experimental observation. These are also important for understanding the materials under investigation.

3. Regarding the authors' reply to question #4: The authors calculated band structure with DFT-HSE. However, the result (Figure R5 and Figure S15) is mis-interpreted. The Fermi level was placed right cross the bottom of conduction band. This is clearly wrong for two reasons. First, how can p-doped SnSe/SnSe₂ have fermi level cross the conduction band? Second, if it was clean SnSe/SnSe₂, then there is no reason for the fermi level crossing a single band only. The result is clearly wrong.

4. Regarding the authors' reply to question #5: The authors claimed that the magnetism is due to Stoner mechanism with exchange interaction strength $J=0.2$ eV. There is no clear description about how they calculated the exchange interaction strength. Critical details are missing. Second, they did not answer the question about how the long-range ferromagnetism is stabilized in 2D.

5. Regarding the authors' reply to question #6: As mentioned in the previous review, it is impossible for the two supercells of SnSe and SnSe₂ to be reasonably matched without significant distortion. The reply from the authors did not address my question, instead it caused more confusion. The additional reference the author provided (Ref. 48 ACS Appl. Mater. Interfaces 10, 15, 12831–12838 (2018)) was about lateral/in-plane interface of SnSe₂-SnSe. But the calculations the authors performed were on the out-of-plane interface between SnSe₂ and SnSe. According to their reply - "Nevertheless, these two materials with distinct lattice constants and crystal symmetries can be connected seamlessly by changing the bond length and rotation angle [48]", did the authors imply they performed calculations of in-plane interface? This is clearly against what they showed in Fig 1a and Supp Fig 24. More puzzling, the in-plane unit cells of SnSe₂ and SnSe have two very different shapes, hexagonal vs rectangular. How can they achieve the reasonable commensurate supercell? The authors should provide a very clear, detailed description about the supercell used in their simulation for reviewers to review and for readers to understand the details. The crystal structures shall also be included in supplementary information.

6. In the revised manuscript, the authors mentioned that the polarization was calculated by "bond-valence method and Debye equation." Details along with references shall be provided. In addition, which code was used in the calculation?

Reviewer #2 (Remarks to the Author):

It seems that the great concerns about the origin of magnetism have been raised by all the referees. Although the authors have made great effort in addressing this issues by taking theoretical explanation (the kinetic exchange mechanism) and some additional experimental measurement, the real observation of magnetic domain and switching is high recommended by magnetic force microscope as they have done on ferroelectricity using PFM.

Reviewer #3 (Remarks to the Author):

In the manuscript entitled "Two-dimensional multiferroic material of metallic p-doped SnSe", 2D p-doped SnSe with the property of ferroelectric and ferromagnetic order in a single material. The authors have been asked to improve the technical aspects of the manuscript in terms of material analysis and ferrimagnetic characterization. The author addressed well and improved the scientific quality of the manuscript. I recommend the manuscript to publish in the journal.

Reviewer #1 (Remarks to the Author):

The authors did not address several questions raised in the previous comments. I cannot recommend it for publication on Nature Communications.

Our response:

We are very thankful for your constructive comments. In this revised manuscript, we have addressed all the questions of your concern. Especially, additionally theoretical calculations are made to further clarify the p-doped feature of SnSe, the MFM measurements are performed to confirm the 2D long-rang ferromagnetic order, and the STEM characterizations are implemented to determine the stacking geometry of SnSe. In addition, these issues of you raised are considered very carefully and addressed point-by-point as follows.

1. Regarding the authors' reply to question #1: For the specific two geometries of antiferroelectric and ferroelectric phase, the antiferroelectric phase is only 3.23 meV higher than ferroelectric phase. Moreover, as the authors mentioned, the supercell was significantly stretched in their calculations. How can the conclusion be safely drawn?

Our response:

We agree with your viewpoint that the energy difference (~ 3.23 meV) between the antiferroelectric and ferroelectric phase is relatively small for 2D p-doped SnSe. During the calculation process, the SnSe/SnSe₂ heterostructure is constructed by intercalating monolayer SnSe₂ into the bilayer SnSe with a distance of ~ 9.7 Å, the large distance between SnSe layers results in the weak interlayer coupling and possibly induces the relatively small energy difference between the antiferroelectric and ferroelectric phase. Even so, this energy difference is still comparable with that of bilayer SnSe (~ 2 meV for AA stacking order) (*npj Comput. Mater.* **8**, 47 (2022)). We have added some discussion in **page 12** by “...*The large distance between SnSe layers in the SnSe/SnSe₂ heterostructure results in a weak interlayer coupling and possibly induces the relatively small energy difference between the antiferroelectric and ferroelectric phase. In spite of this, such an energy difference is still comparable with that of bilayer SnSe (~ 2 meV for AA stacking order)²⁵....*”.

We agree with your viewpoint that it is difficult to reasonably match such two supercells of hexagonal SnSe₂ and rectangular SnSe without the significant distortion. To minimize the lattice mismatch between the stacking layers, the in-plane lattice constants of mainly concerned rectangular SnSe remain unchanged ($a = 4.29$ Å, $b = 4.40$ Å) relative to the bulk counterpart. However, the supercell of hexagonal SnSe₂ ($a = 3.87$ Å, $b = 3.87$ Å) is significantly stretched into a rectangle with the in-plane lattice constants of $a = 3.87$ Å, $b = 6.70$ Å. Accordingly, the SnSe/SnSe₂ heterostructure is constructed by stacking 1×3 supercell of SnSe on 1×2 supercell of SnSe₂. Notably, the lattice stretching of hexagonal TMDCs is also applied in the analogous heterostructure, such as black phosphorous/MoS₂ (*J. Phys. Chem. Lett.* **6**, 2483–2488 (2015)). We have provided the detailed crystal structures of SnSe₂, SnSe, and SnSe/SnSe₂ heterostructure in **Figure R1** (or **Figure S16**) and **Table R1** (or **Table S1**), and also added some discussion in **page 8** by “...*To minimize the lattice mismatch between the stacking layers, a rectangular unit cell of SnSe₂ is constructed. And the SnSe/SnSe₂ heterostructure is built by stacking 1×3 supercell of SnSe on 1×2 supercell of SnSe₂. The optimized in-plane lattice constants of monolayer SnSe₂ are $a = 3.87$ Å, $b = 6.70$ Å, while the corresponding in-plane lattice constants of monolayer SnSe are $a = 4.29$ Å, $b = 4.40$ Å. Notably, the similar lattice stretching of hexagonal TMDCs is also applied in the analogous heterostructure, such as black phosphorous/MoS₂³⁰. The detailed crystal structures of SnSe₂, SnSe, and SnSe/SnSe₂ heterostructure are presented in **Fig. S16** and **Table S1**....*”.

Figure R1 (or Fig. S16). The atomic structure of SnSe/SnSe₂, which is constructed by stacking 1 × 3 supercell of SnSe on 1 × 2 supercell of SnSe₂. To minimize the lattice mismatch between the stacking layers, a rectangular unit cell of SnSe₂ is constructed.

Table R1 (or Table S1). The lattice constants of SnSe and SnSe₂

	a (Å)	b (Å)	γ (°)
SnSe ₂	3.87	6.70	90
SnSe	4.29	4.40	90

Generally, in view of the high polarizability of binary lattice, the distinct antiferroelectric stacking order along the interlayer c -axis is usually formed, when two neighboring monolayers reverse the in-plane dipole orientation along the a -axis. Nevertheless, after intercalating the monolayer SnSe₂ into the bilayer SnSe, the stacking geometry should be changed due to the variation of interlayer coupling (*Nat. Commun.* **9**, 47 (2018)). According to the STEM result in Fig. 2g–i, the AA stacking order is obviously observed for the PVD synthesized 2D p-doped SnSe, possibly due to the local phase segregation of SnSe₂ microdomains. Remarkably, this stacking configuration possesses a stable ferroelectric phase, as has been confirmed by the calculated result in **Fig. S24** and the previous report (*npj Comput. Mater.* **8**, 47 (2022)). We have added some discussion in **page 6** by “...*The AA stacking order of SnSe is obviously observed from the atomic-resolution HAADF-STEM result, where the top layer is precisely matched with the bottom layer in the a - b plane, as described by the atomic structure model in Fig. 2i. Remarkably, this stacking configuration possesses a stable ferroelectric phase, which has been clarified by the calculated result²⁵*”.

We have cited new references in **page 6,12** (Ref. 25) and **page 8** (Ref. 30).

2. Regarding the authors' reply to question #2: The authors did not answer my question "The stacking of SnSe sheets themselves is also not clear." What's the stacking geometry in experiment? As the authors stated in the manuscript, the thickness varies from 3.4 nm to 114 nm. When the SnSe/SnSe₂ layers are formed, how many SnSe layers and SnSe₂ layers are in the final structure? and, what's the stacking geometry of SnSe layers? These questions were raised mainly regarding the experimental observation. These are also important for understanding the materials under investigation.

Our response:

We agree with your viewpoint that the stacking geometry of SnSe is critical importance for exploring its ferroelectric property. According to your suggestion, we have performed the HAADF-STEM measurements of 2D p-doped SnSe along the c -axis to further determine the stacking configuration, with the results shown in **Figure R2** (or **Fig. 2g–i**). Interestingly, the AA stacking order is obviously observed from the atomic-resolution HAADF-STEM result (**Figure**

R2a,b), where the top layer is precisely matched with the bottom layer in the *a-b* plane, as described by the atomic structure model in **Figure R2c**. And such a stacking configuration possesses a stable ferroelectric phase, as has been clarified by the DFT calculated result (*npj Comput. Mater.* **8**, 47 (2022)). Notably, the AA stacking order is also one of the stable configurations for bilayer SnSe, which has been confirmed by the previous calculated results (*Phys. Chem. Chem. Phys.* **20**, 214–220 (2018); *Appl. Surf. Sci.* **512**, 14564 (2020)). We have added some discussion in **page 6** by “...The AA stacking order of SnSe is obviously observed from the atomic-resolution HAADF-STEM result, where the top layer is precisely matched with the bottom layer in the *a-b* plane, as described by the atomic structure model in **Fig. 2i**. Remarkably, this stacking configuration possesses a stable ferroelectric phase, which has been clarified by the calculated result²⁵....”.

*Figure R2 (or Figure 2). (a) Atomic-resolution HAADF-STEM image of a tetragonal SnSe nanosheet. (b) Zoomed-in HAADF-STEM image. (c) Atomic structure model of tetragonal SnSe viewed along the *c*-axis.*

To determine the thicknesses of SnSe₂ and SnSe in the SnSe/SnSe₂ heterostructure (with the thickness of ~16 nm), the XPS depth analysis is thus performed with the result shown in **Figure R3** (or **Fig. S15**). At 0~2 nm depth, the atomic ratio of Sn:Se is over 1:1, indicating that only SnSe exists on the surface of p-doped SnSe. However, at 2~4 nm depth, the atomic ratio of Sn:Se is below 1:1, suggestive of the formation of SnSe₂ microdomains. Interestingly, further increasing the depth from ~4 to ~16 nm, the atomic ratio of Sn:Se is obtained to be over 1:1 again, indicating the disappearance of SnSe₂ microdomains. In terms of the XPS depth analysis result, the thicknesses of SnSe₂ and SnSe are determined to be 2~4 nm and 14~12 nm, respectively, for the SnSe/SnSe₂ heterostructure with a thickness of ~16 nm. Notably, the similar thickness distribution of SnSe₂ microdomains is also demonstrated in the bulk p-doped SnSe (*Nat. Commun.* **9**, 47 (2018)). We have added some discussion in **page 8** by “...The XPS depth analysis was also carried out on SnSe/SnSe₂ heterostructure (with the thickness of ~16 nm) to explore the thickness distribution of SnSe₂ microdomains, and their thicknesses were determined to be 2~4 nm (**Fig. S15**)....”.

Figure R3 (or Fig. S15). The XPS depth analysis of SnSe/SnSe₂ heterostructure with the thickness of ~16 nm. The thicknesses of SnSe₂ and SnSe are determined to be 2~4 nm and 14~12 nm, respectively.

3. Regarding the authors' reply to question #4: The authors calculated band structure with DFT-HSE. However, the result (Figure R5 and Figure S15) is mis-interpreted. The Fermi level was placed right cross the bottom of conduction band. This is clearly wrong for two reasons. First, how can p-doped SnSe/SnSe₂ have fermi level cross the conduction band? Second, if it was clean SnSe/SnSe₂, then there is no reason for the fermi level crossing a single band only. The result is clearly wrong.

Our response:

We agree with your viewpoint that the band structure calculated by using DFT-HSE method shows the discrepancy with the experimental result due to the neglect of interfacial charge transfer effect on the position of Fermi level. We have recalculated the band structure and reset the location of Fermi level of SnSe/SnSe₂, with the result shown in **Figure R4** (or **Fig. S17**). Notably, according to the band structure calculated by the DFT-PBE method, it is impossible for conducting along the X-R-Y line, and thus we have recalculated the band structure of SnSe₂/SnS along the X- Γ -Y line by using the DFT-HSE method. Interestingly, the Fermi level of SnSe/SnSe₂ heterostructure is crossed by the valence band comparing with that of intrinsic monolayer SnSe (*Phys. Rev. B* **96**, 165118 (2017); *Phys. Rev. Appl.* **13**, 014042 (2020)), highly suggestive of its p-doped feature. Notably, similar calculation result and p-doped property are also demonstrated in the bulk SnSe with local phase segregation of SnSe₂ microdomains (*Nat. Commun.* **9**, 47 (2018)). In addition, according to the Hall measurement result in **Fig. S10**, the carrier concentration of p-doped SnSe is increased with elevating the temperature, which should further down shift the Fermi level, as has been confirmed by the ARPES, KPFM, and DFT calculated results (*Nat. Commun.* **9**, 47 (2018); *Phys. Rev. B* **97**, 245202 (2018)), and thus the p-doped feature is further convinced.

Figure R4 (or Fig. S17). Calculated the band structure of SnSe/SnSe₂ by using DFT method with the HSE function. The Fermi level is crossed by several bands, indicative of its metallic feature.

The p-doped mechanism in SnSe by intercalating SnSe₂ microdomains can be well understood through the interfacial charge transfer. As sketched in **Fig. 3g**, the large VBM difference results in the interfacial charge transfer from SnSe to SnSe₂, and thus an effective p-doping in SnSe. In a word, according to the Hall measurement result (**Fig. S10**) and the previous report (*Nat. Commun.* **9**, 47 (2018)), the p-doped property is credible. According to your comment, we have revised the band structure of SnSe₂/SnSe in **Fig. S17**.

4. Regarding the authors' reply to question #5: The authors claimed that the magnetism is due to Stoner mechanism with exchange interaction strength $J=0.2$ eV. There is no clear description about how they calculated the exchange interaction strength. Critical details are missing. Second, they did not answer the question about how the long-range ferromagnetism is stabilized in 2D.

Our response:

According to your suggestion, the detailed calculation process about the exchange interaction strength (J) of p-doped SnSe is added in the method section (**page 16**) by “...*Exchange interaction strength calculation. The exchange interaction strength (J) was calculated by using the four-state method⁵⁵ and then the magnetic states were constructed to be $E_1 (\uparrow, \uparrow) = -468.32202313$ eV, $E_2 (\uparrow, \downarrow) = -468.42250830$ eV, $E_3 (\downarrow, \uparrow) = -468.42250830$ eV, and $E_4 (\downarrow, \downarrow) = -468.32202313$ eV, respectively. Accordingly, the J value can be calculated to be ~ 0.2 eV, by using the formula of $J = (E_1 + E_4 - E_2 - E_3)/4S^2$. The spin-polarized calculations were performed using the VASP based on PAW method. Electron-ion interactions were described by the standard PAW potentials. The plane wave basis was set as 400 eV to expand the smooth part of wave functions. For the electron-electron exchange and correlation interactions, the functional parametrized by PBE (a form of GGA) was used throughout. The van der Waals interaction was described via the DFT-D3BJ method. During the geometry optimizations, the bottom atoms were fixed at the bulk form. And the Brillouin-zone integrations were conducted by using Monkhorst-Pack grids of special points with a separation of 0.06 \AA^{-1} . The convergence criterion for the electronic self-consistent loop was set as 10^{-4} eV. The atomic structures were optimized until the residual forces were below 0.03 eV \AA^{-1}”.*

We have calculated the exchange interaction strength ($J \sim 0.20$ eV) of 2D p-doped SnSe, which is comparable with those of 2D ferromagnetic FeTe (0.30~0.40 eV) (*Nat. Commun.* **11**, 3729 (2020)) and ferromagnetic transition-metal halides (0.20~0.57 eV) (*Phys. Rev. Mater.* **5**, 034001 (2021)). Such a large exchange interaction strength in 2D p-doped SnSe suggests the strong interaction and coupling between the magnetic moments, which is favorable for stabilizing the long-range ferromagnetism. We have added some discussion in **page 10** by “...*The obtained value of J is comparable with those of 2D ferromagnetic FeTe (0.30~0.40 eV)³⁹ and ferromagnetic transition-metal halides (0.20~0.57 eV)⁴⁰. Such a large exchange interaction strength in 2D p-doped SnSe indicates the strong interaction and coupling between the magnetic moments, which is favorable for stabilizing the long-range ferromagnetism....*”. Besides, according to the temperature-dependent longitudinal resistance (**Fig. 3c**) and magnetoresistance (**Fig. S18**) results, the long-range Ruderman-Kittel-Kasuya-Yosida (RKKY) interaction is proved to be dominant, which also stabilizes the long-range ferromagnetic order in 2D p-doped SnSe. We have also added some discussion in **page 11** by “...*Moreover, the reduced longitudinal resistance with decreasing the temperature (Fig. 3c) and the appearance of positive magnetoresistance throughout the whole temperature range (Fig. S18) suggest the robust long-range Ruderman-Kittel-Kasuya-Yosida (RKKY) interaction, which is also favorable for stabilizing the long-range ferromagnetic order in 2D p-doped SnSe....*”.

In order to further confirm the long-range ferromagnetic order and determine the magnetic domain, the MFM measurements are performed on PVD synthesized p-doped SnSe with different thicknesses at room-temperature (**Figure R5** or **Fig. S21**). Interestingly, the phase deviation between p-doped SnSe and nonmagnetic SiO₂/Si substrate is obviously observed, indicating its ferromagnetic order, and the similar results are also demonstrated in monolayer V-doped WSe₂ (*Adv. Mater.* **34**, 2106551 (2022)) and monolayer VSe₂ (*Adv. Mater.* **31**, 1903779 (2019)). Notably, the single magnetic domain structure is obtained for p-doped SnSe, which is similar to 2D ferromagnetism Fe₃GaTe₂ (*Nat. Commun.* **13**, 5067 (2022)). Therefore, from the experimental point of view, the SQUID (**Fig. 4a-d**) and MFM (**Fig. S21**) measurement results collectively confirm the long-range ferromagnetic order in PVD synthesized 2D p-doped SnSe.

We have cited new references in **page 10** (Ref. 39,40) and **page 16** (Ref. 55).

Figure R5 (or Fig. S21). MFM measurements of PVD synthesized *p*-doped SnSe with different thicknesses at room-temperature. (a) AFM image and corresponding height profile analysis of a *p*-doped SnSe nanosheet with the thickness of ~ 33.5 nm. (b) Corresponding MFM phase image. The phase deviation between *p*-doped SnSe and nonmagnetic SiO₂/Si substrate indicates the ferromagnetic order of *p*-doped SnSe. The single magnetic domain structure is obtained for *p*-doped SnSe. (c) AFM image and corresponding height profile analysis of a *p*-doped SnSe nanosheet with the thickness of ~ 44.0 nm. (d–f) Corresponding MFM phase images captured from different DC voltages.

5. Regarding the authors' reply to question #6: As mentioned in the previous review, it is impossible for the two supercells of SnSe and SnSe₂ to be reasonably matched without significant distortion. The reply from the authors did not address my question, instead it caused more confusion. The additional reference the author provided (Ref. 48 ACS Appl. Mater. Interfaces 10, 15, 12831–12838 (2018)) was about lateral/in-plane interface of SnSe₂-SnSe. But the calculations the authors performed were on the out-of-plane interface between SnSe₂ and SnSe. According to their reply - "Nevertheless, these two materials with distinct lattice constants and crystal symmetries can be connected seamlessly by changing the bond length and rotation angle [48]", did the authors imply they performed calculations of in-plane interface? This is clearly against what they showed in Fig 1a and Supp Fig 24. More puzzling, the in-plane unit cells of SnSe₂ and SnSe have two very different shapes, hexagonal vs rectangular. How can they achieve the reasonable commensurate supercell? The authors should provide a very clear, detailed description about the supercell used in their simulation for reviewers to review and for readers to understand the details. The crystal structures shall also be included in supplementary information.

Our response:

We agree with your viewpoint that it is difficult to reasonably match such two supercells of hexagonal SnSe₂ and rectangular SnSe without the significant distortion. To minimize the lattice mismatch between the stacking layers, the in-plane lattice constants of mainly concerned rectangular SnSe remain unchanged ($a = 4.29$ Å, $b = 4.40$ Å) relative to the bulk counterpart. However, the supercell of hexagonal SnSe₂ ($a = 3.87$ Å, $b = 3.87$ Å) is significantly stretched into a rectangle with the in-plane lattice constants of $a = 3.87$ Å, $b = 6.70$ Å. Accordingly, the SnSe/SnSe₂ heterostructure is constructed by stacking 1×3 supercell of SnSe on 1×2 supercell of SnSe₂. Notably, the similar lattice stretching of hexagonal TMDCs is also applied in the analogous heterostructure, such as black phosphorous/MoS₂ (*J. Phys. Chem. Lett.* **6**, 2483–2488 (2015)). According to your suggestion, we have provided the detailed crystal structures of SnSe₂, SnSe, and SnSe/SnSe₂ heterostructure in **Figure R6** (or **Fig. S16**) and **Table R2** (or **Table S1**), and also added the detailed description regarding the construction of SnSe/SnSe₂ heterostructure in

page 8 by “...To minimize the lattice mismatch between the stacking layers, a rectangular unit cell of SnSe₂ is constructed. And the SnSe/SnSe₂ heterostructure is built by stacking 1 × 3 supercell of SnSe on 1 × 2 supercell of SnSe₂. The optimized in-plane lattice constants of monolayer SnSe₂ are a = 3.87 Å, b = 6.70 Å, while the corresponding in-plane lattice constants of monolayer SnSe are a = 4.29 Å, b = 4.40 Å. Notably, the similar lattice stretching of hexagonal TMDCs is also applied in the analogous heterostructure, such as black phosphorous/MoS₂³⁰. The detailed crystal structures of SnSe₂, SnSe, and SnSe/SnSe₂ heterostructure are presented in Fig. S16 and Table S1...”.

Figure R6 (or Fig. S16). The atomic structure of SnSe/SnSe₂, which is constructed by stacking 1 × 3 supercell of SnSe on 1 × 2 supercell of SnSe₂. To minimize the lattice mismatch between the stacking layers, a rectangular unit cell of SnSe₂ is constructed.

Table R2 (or Table S1). The lattice constants of SnSe and SnSe₂

	a (Å)	b (Å)	γ (°)
SnSe ₂	3.87	6.67	90
SnSe	4.29	4.40	90

We admit that the description of SnSe₂-SnSe in-plane heterostructure is inappropriate for understanding the construction process of SnSe/SnSe₂ vertical heterostructure, and we have removed the related discussion in page 15.

6. In the revised manuscript, the authors mentioned that the polarization was calculated by "bond-valence method and Debye equation." Details along with references shall be provided. In addition, which code was used in the calculation?

Our response:

According to your kind suggestion, we have provided the details about the bond-valence method and Debye equation along with references in page 16 by “...Dipole moment calculation. The Debye equation, which was defined as $\mu = neR$ (where μ was the net dipole moment in Debye (10^{-18} esu cm), n was the total number of electrons, e was the charge on an electron (-4.8×10^{-10} esu), and R was the difference in cm (between the “centroids” of positive and negative charges)), was used to calculate the dipole moment of 2D p-doped SnSe⁵⁶. The distribution of electrons on Sn/Se atoms was estimated by the bond valence method (as described by the following formula of $S_i = \exp[(R_0 - R_i)/B]$, where R_0 was an empirical constant (Sn^(II)-Se, $R_0 = 2.476$ Å; Sn^(IV)-Se, $R_0 = 2.524$ Å), R_i was the length of bond “i” in Å, and $B = 0.37$ Å)⁵⁷. The refined atomic coordinates and selected bond distances were shown in Table S3 and Table S4, respectively....”.

During the DFT calculation process, the VASP code is applied, while the crystal structure analysis of p-doped SnSe

is performed by using the VESTA code.

Table R3 (or Table S3). The refined atomic coordinates of SnSe and SnSe/SnSe₂

SnSe				SnSe/SnSe ₂			
Atom	x	y	z	Atom	x	y	z
Sn1	0.9625	0.1465	0.3760	Sn1	0.98806	0.15033	0.37842
Sn2	0.4729	0.3098	0.4547	Sn2	0.48818	0.31479	0.46204
Sn3	0.9639	0.4794	0.3760	Sn3	0.98819	0.48503	0.38458
Sn4	0.4751	0.6431	0.4544	Sn4	0.48795	0.65147	0.46216
Sn5	0.9647	0.8128	0.3757	Sn5	0.98876	0.81461	0.37812
Sn6	0.4732	0.9763	0.4545	Sn6	0.48813	0.98514	0.45785
Sn7	0.2755	0.1385	0.0308	Sn7	0.99052	0.19134	0.24364
Sn8	0.7706	0.3044	0.1116	Sn8	0.49442	0.44078	0.24361
Sn9	0.2771	0.4715	0.0303	Sn9	0.98473	0.69343	0.24624
Sn10	0.7705	0.6379	0.1113	Sn10	0.49165	0.94319	0.24752
Sn11	0.2775	0.8052	0.0302	Sn11	0.21672	0.15078	0.02906
Sn12	0.7708	0.9708	0.1114	Sn12	0.71973	0.31766	0.10789
Sn13	0.8473	0.1338	0.1990	Sn13	0.21872	0.48400	0.02859
Sn14	0.3513	0.3052	0.2797	Sn14	0.71414	0.65096	0.10793
Sn15	0.8481	0.4677	0.1989	Sn15	0.22038	0.81708	0.02806
Sn16	0.3515	0.6383	0.2798	Sn16	0.72069	0.98321	0.10884
Sn17	0.8507	0.8005	0.1988	Se1	0.48804	0.33749	0.38567
Sn18	0.3530	0.9715	0.2798	Se2	0.98766	0.16837	0.45706
Se1	0.4721	0.3287	0.3772	Se3	0.48743	0.66481	0.38485
Se2	0.9727	0.1624	0.4539	Se4	0.98654	0.50460	0.46251
Se3	0.4739	0.6618	0.3768	Se5	0.48770	0.00242	0.38101
Se4	0.9736	0.4958	0.4539	Se6	0.98887	0.83798	0.45549
Se5	0.4719	-0.0043	0.3770	Se7	0.99234	0.02629	0.29057
Se6	0.9730	0.8288	0.4535	Se8	0.49135	0.10684	0.20087
Se7	0.7782	0.3246	0.0341	Se9	0.49116	0.2719	0.28719
Se8	0.2797	0.1576	0.1077	Se10	0.99368	0.36121	0.19938
Se9	0.7789	0.6583	0.0339	Se11	0.98926	0.52054	0.28828
Se10	0.2801	0.4910	0.1072	Se12	0.49181	0.61413	0.20181
Se11	0.7776	-0.0085	0.0341	Se13	0.4856	0.77314	0.2908
Se12	0.2797	0.8242	0.1071	Se14	0.98198	0.86323	0.20356
Se13	0.3392	0.3221	0.2019	Se15	0.72086	0.33676	0.02974
Se14	0.8413	0.1553	0.2773	Se16	0.21614	0.16854	0.10582
Se15	0.3407	0.6553	0.2020	Se17	0.71887	0.66984	0.02978
Se16	0.8415	0.4885	0.2774	Se18	0.21541	0.50174	0.1055
Se17	0.3392	0.9882	0.2019	Se19	0.71795	0.00376	0.03109
Se18	0.8421	0.8217	0.2772	Se20	0.2231	0.83381	0.10486

Table R4 (or Table S4). The bond distances (Å) of SnSe and SnSe/SnSe₂

SnSe		SnSe/SnSe ₂	
Sn1-Se2	2.76157	Sn1-Se2	2.78958
Sn1-Se5	2.95772	Sn1-Se5	2.90228
Sn2-Se1	2.75045	Sn2-Se1	2.71576
Sn2-Se2	2.89695	Sn2-Se2	2.89246
Sn3-Se1	2.95312	Sn3-Se1	2.89847
Sn3-Se4	2.76205	Sn3-Se4	2.76640
Sn4-Se3	2.75373	Sn4-Se3	2.73805
Sn4-Se4	2.89185	Sn4-Se4	2.88829
Sn5-Se3	2.95891	Sn5-Se3	2.92436
Sn5-Se6	2.75803	Sn5-Se6	2.75187
Sn6-Se5	2.75106	Sn6-Se5	2.72534
Sn6-Se6	2.89791	Sn6-Se6	2.89887
Sn7-Se8	2.72963	Sn7-Se7	2.74014
Sn7-Se11	2.88958	Sn7-Se8	2.84915
Sn8-Se7	2.75225	Sn7-Se9	2.84873
Sn8-Se8	2.92444	Sn7-Se10	2.73611
Sn9-Se7	2.89263	Sn8-Se9	2.71166
Sn9-Se10	2.73010	Sn8-Se10	2.85250
Sn10-Se9	2.74904	Sn8-Se11	2.84781
Sn10-Se10	2.86627	Sn8-Se12	2.72587
Sn11-Se9	2.90041	Sn9-Se11	2.72542
Sn11-Se12	2.72947	Sn9-Se12	2.88032
Sn12-Se11	2.74583	Sn9-Se13	2.85917
Sn12-Se12	2.86616	Sn9-Se14	2.70379
Sn13-Se14	2.78202	Sn10-Se7	2.85214
Sn13-Se17	2.85757	Sn10-Se8	2.71942
Sn14-Se14	2.88963	Sn10-Se13	2.71851
Sn14-Se13	2.75923	Sn10-Se14	2.82050
Sn15-Se13	2.85517	Sn11-Se16	2.72306
Sn15-Se16	2.78815	Sn11-Se19	2.89887
Sn16-Se15	2.75924	Sn12-Se15	2.77357
Sn16-Se16	2.88873	Sn12-Se16	2.90237
Sn17-Se15	2.84840	Sn13-Se15	2.90312
Sn17-Se18	2.78526	Sn13-Se18	2.72836
Sn18-Se17	2.76268	Sn14-Se17	2.77337
Sn18-Se18	2.88614	Sn14-Se18	2.91895
		Sn15-Se17	2.89187
		Sn15-Se20	2.72336
		Sn16-Se19	2.76133
		Sn16-Se20	2.92628

We have cited new references in **page 16** (Ref. 56,57).

Again, we would like to thank your thoughtful comments and suggestion, which we think have helped to greatly improve the readability and clarity of our manuscript.

Reviewer #2 (Remarks to the Author):

It seems that the great concerns about the origin of magnetism have been raised by all the referees. Although the authors have made great effort in addressing these issues by taking theoretical explanation (the kinetic exchange mechanism) and some additional experimental measurement, the real observation of magnetic domain and switching is high recommended by magnetic force microscope as they have done on ferroelectricity using PFM.

Our response:

We are very thankful for your constructive comment.

We agree with your viewpoint that the direct observation of magnetic domain and switching behavior is critical importance, and then the MFM measurements are performed on PVD synthesized p-doped SnSe with different thicknesses at room-temperature (**Figure R7** or **Fig. S21**). Interestingly, the phase deviation between p-doped SnSe and nonmagnetic SiO₂/Si substrate is obviously observed, highly suggestive of its ferromagnetic order, and the similar results are also demonstrated in monolayer V-doped WSe₂ (*Adv. Mater.* **34**, 2106551 (2022)) and monolayer VSe₂ (*Adv. Mater.* **31**, 1903779 (2019)). Notably, the single magnetic domain structure is obtained for p-doped SnSe, which is similar to 2D ferromagnetism Fe₃GaTe₂ (*Nat. Commun.* **13**, 5067 (2022)). In view of the relatively weak saturation magnetization and coercivity, as well as the ferrimagnetic feature of 2D p-doped SnSe, the phase deviation is small but still comparable with that of monolayer VSe₂ (*Adv. Mater.* **31**, 1903779 (2019)).

Figure R7 (or Fig. S21). MFM measurements of PVD synthesized p-doped SnSe with different thicknesses at room-temperature. (a) AFM image and corresponding height profile analysis of a p-doped SnSe nanosheet with the thickness of ~33.5 nm. (b) Corresponding MFM phase image. The phase deviation between p-doped SnSe and nonmagnetic SiO₂/Si substrate indicates the ferromagnetic order of p-doped SnSe. The single magnetic domain structure is obtained for p-doped SnSe. (c) AFM image and corresponding height profile analysis of a p-doped SnSe nanosheet with the thickness of ~44.0 nm. (d–f) Corresponding MFM phase images captured from different DC voltages.

In addition, the switching behavior is also explored by applying the DC voltage (± 10 V) during the MFM measurements. Remarkably, the phase deviation between p-doped SnSe and nonmagnetic SiO₂/Si substrate is slightly increased from ~ 0.065 to $\sim 0.070^\circ$, after introducing +10 V voltage, nevertheless, the switching behavior between the ferromagnetic and paramagnetic order is not observed, possibly due to the weak magnetoelectric coupling in 2D p-doped SnSe. Even so, this work still presents the first report regarding the coexistence of ferrimagnetism and ferroelectricity in the single phase of 2D p-doped SnSe, which provides a solid foundation for exploring the exotic quantum phenomena and exploiting multifunctional applications in nanoelectronics and thermoelectrics. We have added some discussion in **page 10** by “...*To further determine the magnetic domain and switching behavior, the magnetic force microscope (MFM) measurements were then performed on the transferred p-doped SnSe with different thicknesses at room-temperature. Interestingly, the phase deviation between p-doped SnSe and nonmagnetic SiO₂/Si substrate is obviously observed, indicating its ferromagnetic order (Fig. S21), the similar phenomena are also demonstrated in monolayer V-doped WSe₂³⁵ and monolayer VSe₂³⁶. Notably, the single magnetic domain structure is obtained for p-doped SnSe, which is similar to 2D ferromagnetism Fe₃GaTe₂³⁷. However, the switching behavior between the ferromagnetic and paramagnetic order is not observed, possibly due to the weak magnetoelectric coupling in 2D p-doped SnSe....*”.

We have cited new references in **page 10** (Ref. 35,36,37).

Again, we would like to thank your thoughtful comments and suggestion, which we think have helped to greatly improve the readability and clarity of our manuscript.

Reviewer #3 (Remarks to the Author):

In the manuscript entitled “Two-dimensional multiferroic material of metallic p-doped SnSe”, 2D p-doped SnSe with the property of ferroelectric and ferromagnetic order in a single material. The authors have been asked to improve the technical aspects of the manuscript in terms of material analysis and ferrimagnetic characterization. The author addressed well and improved the scientific quality of the manuscript. I recommend the manuscript to publish in the journal.

Our response:

We are very thankful for your recommendation.

Again, we would like to thank your thoughtful comments and suggestion, which we think have helped to greatly improve the readability and clarity of our manuscript.

REVIEWERS' COMMENTS

Reviewer #2 (Remarks to the Author):

The authors have made great effort in addressing the concerns from the referees and improving the quality of the manuscript. I recommend this work to be accepted for publication.

Reviewer #2 (Remarks to the Author):

The authors have made great effort in addressing the concerns from the referees and improving the quality of the manuscript. I recommend this work to be accepted for publication.

Our response:

We would like to thank you for reviewing our paper, we appreciate your insightful comments on our research.

Again, we would like to thank the other reviewers' comments and suggestion, which we think have helped to greatly improve the readability and clarity of our manuscript.